# Soil–vegetation–water interactions controlling solute flow and chemical weathering in volcanic ash soils of the high Andes

**Sebastián Páez-Bimos[1,2], Armando Molina[3], Marlon Calispa[3,4], Pierre Delmelle[4], Braulio Lahuatte[5], Marcos Villacís[1], Teresa Muñoz[6], and Veerle Vanacker[2]**

[1]Departamento de Ingeniería Civil y Ambiental & Centro de Investigaciones y Estudios en Ingeniería de los Recursos Hídricos (CIERHI), Facultad de Ingeniería Civil y Ambiental, Escuela Politécnica Nacional, Quito, 170525, Ecuador
[2]Earth and Climate Research, Earth and Life Institute, UCLouvain, Louvain-la-Neuve, 1348, Belgium
[3]Programa para el Manejo de Agua y Suelo (PROMAS), Facultad de Ingeniería Civil,
Universidad de Cuenca, Cuenca, 010203, Ecuador
[4]Environmental Sciences, Earth and Life Institute, UCLouvain, Louvain-la-Neuve, 1348, Belgium
[5]Fondo para la Protección del Agua (FONAG), 170509, Quito, Ecuador
[6]Empresa Pública Metropolitana de Agua Potable y Saneamiento (EPMAPS), Quito, 170519, Ecuador

**Correspondence:** Sebastián Páez-Bimos (carlos.paezb@epn.edu.ec) and Veerle Vanacker (veerle.vanacker@uclouvain.be)

Received: 10 August 2022 – Discussion started: 9 September 2022
Accepted: 23 December 2022 – Published:

**Abstract.** TS1 TS2 Vegetation plays a key role in the hydrological and biogeochemical cycles. It can influence soil water fluxes and transport, which are critical for chemical weathering and soil development. In this study, we investigated soil water balance and solute fluxes in two soil profiles with different vegetation types (cushion-forming plants vs. tussock grasses) in the high Ecuadorian Andes by measuring soil water content, flux, and solute concentrations and by modeling soil hydrology. We also analyzed the role of soil water balance in soil chemical weathering. The influence of vegetation on soil water balance and solute fluxes is restricted to the A horizon. Evapotranspiration is 1.7 times higher and deep drainage 3 times lower under cushion-forming plants than under tussock grass. Likewise, cushions transmit about 2-fold less water from the A to lower horizons. This is attributed to the higher soil water retention and saturated hydraulic conductivity associated with a shallower and coarser root system. Under cushion-forming plants, dissolved organic carbon (DOC) and metals (Al, Fe) are mobilized in the A horizon. Solute fluxes that can be related to plant nutrient uptake (Mg, Ca, K) decline with depth, as expected from biocycling of plant nutrients. Dissolved silica and bicarbonate are minimally influenced by vegetation and represent the largest contributions of solute fluxes. Soil chemical weathering is higher and constant with depth below tussock grasses but lower and declining with depth under cushion-forming plants. This difference in soil weathering is attributed mainly to the water fluxes. Our findings reveal that vegetation can modify soil properties in the uppermost horizon, altering the water balance, solute fluxes, and chemical weathering throughout the soil profile.

## 1 Introduction

Soil hydrology regulates the chemical weathering of primary minerals in the regolith (Maher, 2010; Maher and Chamberlain, 2014; Velbel, 1993). Soil water flux and transport depend on soil water content (Rodriguez-Iturbe, 2000), and both are critical for chemical weathering and soil development (Brantley et al., 2011; Calabrese and Porporato, 2020). For a given availability of reactive weathering products, slow water flux and transport within the soil mantle facilitate the build-up of solute concentrations (up to saturation) and eventually the formation of secondary solid phases (Pope, 2015). In contrast, high soil water flux (e.g., during hydrological events) can flush out water-soluble products from the vadose zone, resulting in a reduction of concentrations of weathering products in the soil (Berner and Berner, 2012; Perdrial et al., 2015). In well-drained soil systems (e.g., fluid residence

times between 5 d and 10 years), chemical weathering rates are proportional to water fluxes (Berner, 1978; Lasaga et al., 1994; Maher, 2010).

The influence of soil hydrology on chemical weathering rates has mostly been assessed through proxy variables, like soil water availability or soil moisture (Daly and Porporato, 2005; Moore et al., 2015; White et al., 2005). Soil water availability can be approached using the ratio of mean annual rainfall to potential evapotranspiration, as illustrated in studies of soil development along climatic gradients (Chadwick et al., 2003; Dixon et al., 2016; Schoonejans et al., 2016) and at the global scale (Calabrese and Porporato, 2020). While these investigations have recognized the importance of water flux and transport in soil weathering, only a few of them have directly assessed how soil water fluxes (Clow and Drever, 1996; Maher, 2010; White et al., 2009) and hydrological processes, like infiltration and storage (Cipolla et al., 2021; García-Gamero et al., 2022), control weathering processes. Moreover, the effect of the water balance on soil weathering can be elusive due to the potential influence of less-explored co-evolving soil formation factors, notably lithological and climatic settings (Schoonejans et al., 2016).

Soil weathering can be assessed from the solid and solute chemical distributions in the soil mantle. Changes in the solid-phase soil mass derived from elemental mass balances are measures of long-term weathering (White, 1995; White et al., 1998, 2009), whereas contemporary solute fluxes reflect weathering rates at short timescales (White and Buss, 2014; White, 1995). Chemical weathering is conditioned by the intrinsic properties of the soil particles (e.g., porosity, soil particle surface area, mineralogy) and the soil solution (e.g., solution pH, conductivity, temperature) (Anderson et al., 2007). Soil weathering processes vary with depth: chemical weathering is more pronounced near the surface (A and E horizons) and decreases with depth through the B horizon to the less-weathered C horizon (White, 1995). The depth gradient in weathering extent is associated with the solution pH, dissolved Al, organic and carbonic acids influencing dissolution reactions, and the hydraulic conductivity that affects water flux and transport (Anderson et al., 2007). The water residence time plays a critical role in the depth variation of the soil weathering extent, as it increases with depth in the soil, leading to a longer time for reaction between water and the surfaces of soil particles (Pope, 2015). Vegetation can directly control soil weathering depth by altering the composition of the soil solution, but also indirectly by influencing soil hydrology (Kelly et al., 1998).

Vegetation plays a key role in the hydrological cycle at different spatial and temporal scales by extracting water from the soil and influencing water pathways and fluxes (Brantley et al., 2017; Drever, 1994; Kelly et al., 1998; Moore et al., 2015). The soil water availability and seasonal water balance can, in turn, determine the distribution of vegetation types (Tromp-van Meerveld and McDonnell, 2006; Berghuijs et al., 2014). Vegetation development can co-evolve with soil weathering extent by vegetation adapting to and transforming its environment through biogeochemical weathering (Sivapalan, 2018). The influence of soil type and properties can overrule the effects of vegetation type on dynamic soil water storage change and residence time under conditions of low precipitation seasonality and low evaporation (Geris et al., 2015). Conversely, soil structure and hydraulic properties associated with vegetation (e.g., distance from trees) can control the spatial pattern of soil water content (Metzger et al., 2017). Moreover, the presence and architecture of root systems can increase soil porosity and saturated hydraulic conductivity affecting water infiltration (Jiang et al., 2018) and explain the variability in evapotranspiration under a given climatic condition (Hunt, 2021).

Vegetation can directly influence biogeochemical processes and facilitate soil weathering in several ways. These include (i) root and microbial respiration, increasing concentrations of carbon dioxide in the soil and thus lowering soil solution pH, (ii) root penetration, enhancing pathways for subsurface flow, (iii) production of organic acids and compounds from the decay of organic matter or root exudation, resulting in lower pH and in chelates that mobilize soluble metal complexes and alter nutrient exchanges, (iv) uptake of water and solutes in the rhizosphere, resulting in changes in ion concentrations in soil solutions, and (v) cycling nutrients (e.g., Ca, Mg, K) through litter and roots, which can result in higher concentrations in soil solutions in the upper horizons (Brantley et al., 2012; Hinsinger et al., 2006; Kelly et al., 1998; Pope, 2015). Likewise, biota effects on weathering can also be present in chemical gradients (e.g., pH, solute concentrations) along with soil depth, even at millimetric scales (Chorover et al., 2007).

The effect of soil hydrology on chemical weathering has typically been studied indirectly through meteorological variables, such as studies using long-term water balances based on the Budyko framework (e.g., Calabrese and Porporato, 2020; Hunt, 2021). While such indirect assessments are useful for large-scale studies, they fail in capturing the variability in soil properties, topography, and vegetation patterns that may exist at small spatial scales (Calabrese et al., 2022; Li et al., 2013; Sullivan et al., 2022). Here, we address this research gap by taking advantage of the mosaic-like distribution of vegetation types in the high Andes ecosystem, changing over short distances and allowing other factors (i.e., climate, geology, soil age, and topography) to remain constant (Molina et al., 2019). The main research questions motivating this study are the following. (i) What are the effects of vegetation type and the associated soil properties on soil water balance? (ii) To what extent do soil–vegetation associations alter solute fluxes? (iii) How does vegetation alter contemporary soil weathering through the soil water balance? To analyze vegetation–soil associations in relation to the soil water balance, we used the HYDRUS-1D model to simulate soil hydrological processes, including evapotranspiration, deep drainage, and soil water storage. Simulated soil

moisture and water fluxes at soil horizons were calibrated and validated with independent field measurements. To analyze the influence of the infiltrated water fluxes on soil chemical weathering, we sampled soil solutions at biweekly intervals, and their compositions served to estimate solute fluxes. Overall, this study assesses the influence of vegetation type and the associated soil properties on soil water balance, solute fluxes, and contemporary soil chemical weathering at the soil profile scale. Given that vegetation patterns in the high Andes are subject to rapid anthropogenic and/or climate change (Molina et al., 2015; Vanacker et al., 2018), this study also contributes to assessing the potential impact of vegetation change on soil hydrophysical and chemical properties, soil water and nutrient balance, and leaching of soil solutes.

## 2  Páramo ecosystem

The high Andean ecosystem, known as páramo, is a cold and humid neotropical alpine region with high solar radiation and low-intensity rainfall. It is situated between forest and snow lines and is characterized by soils of volcanic origin (Aparecido et al., 2018) covered by a highly diverse mosaic of endemic plants adapted to extreme climatic conditions (Myers et al., 2000; Körner, 2003). Soils are characterized by high porosity, high organic matter content, low bulk density and high hydraulic conductivity. Subsurface water flow is dominant (Correa et al., 2017; Mosquera et al., 2022), and overland rainfall runoff is only reported in anthropogenically degraded páramo soils (Harden, 2006). Chemical weathering has been reported as exceeding physical erosion in páramo catchments (Tenorio et al., 2018), similar to what was observed in other alpine environments (Dixon and Thorn, 2005). In the total solute load of páramo stream waters, the fluxes of bicarbonate, dissolved silica and dissolved organic carbon are dominant (Arízaga-Idrovo et al., 2022; Tenorio et al., 2018).

Previous work mostly focused on vegetation effects on individual components of the soil water balance, like interception, evapotranspiration, or soil weathering. Carrillo-Rojas et al. (2019) showed that the actual evapotranspiration can represent half of the annual rainfall in tussock grasslands, and Ochoa-Sánchez et al. (2018) pointed to the importance of vegetation interception, which can vary between 10 % and 100 %, depending on the total rainfall. Significant differences in soil chemical weathering were associated with vegetation patterns (Molina et al., 2019), but it is not yet clear how vegetation can influence contemporary weathering rates through its effect on soil water fluxes and transport. This study contributes to filling this knowledge gap by examining the differences in soil hydrology and chemical weathering rates in two pedons with different vegetation cover located in summit topographic positions. The soil profiles covered by tussock grass and cushion-forming plants consist of polyge-

netic, young volcanic ash soil and are located in the northern Ecuadorian páramo.

## 3  Materials and methods

### 3.1  Study sites

The study sites are located on the western slopes of the Antisana volcano within the Antisana's Water Conservation Area (Antisana's WCA; Fig. 1b). The area is managed by the Fondo de Protección del Agua (FONAG), and since 2011, all anthropogenic activities and extensive grazing have been prohibited. Two soil profiles with distinct vegetation types, i.e., cushion-forming plants (CU-UR) and tussock grass (TU-UP), were excavated at the summit topographic position (Fig. 1a). Due to the location of the soil profiles at the crest of a headwater subcatchment in the Jatunhuayco watershed and their low slope gradients ($\leq 6.5$ %), we posit that vertical downward water flow is dominant in the soil profiles. The mean annual meteorological variables as recorded in 2019–2020 at the JTU_AWS station (Fig. 1a) are summarized as follows: rainfall $723.3 \pm 7.4$ mm, air temperature $4.3 \pm 0.5$ °C, incoming shortwave radiation $169.2 \pm 9.1$ W m$^{-2}$, relative humidity $92.9 \pm 0.9$ %, wind speed $3.4 \pm 0.3$ m s$^{-1}$, and predominant wind direction from east to west. Vegetation species on the TU-UP and CU-UR profiles are dominated by *Calamagrostis intermedia* and *Azorella pedunculata*, respectively. Rooting depth is 70 cm at TU-UP and 30 cm at CU-UR. The soils in the upper 1 m are polygenic vitric Andosols (Calispa et al., 2021), developed from Holocene ash depositions (Hall et al., 2017). Four horizons have been identified: the upper organic-rich A horizon on top of a buried 2A horizon and below two mineral horizons 2BC and 3BC (Table 1). These horizons are characterized by a decreasing gradient with depth of organic carbon, hydraulic saturated conductivity, and water retention (Páez-Bimos et al., 2022). The recent soils are developed on top of a $\sim 27$ m-thick sequence of paleosols and tephra layers that overlay scoria-rich layers and glaciofluvial sediments (Hall et al., 2017).

### 3.2  Soil hydrophysical and chemical properties

Undisturbed soil samples (100 cm$^3$) were collected in duplicate (next to each other) in nine vertical positions (15, 25, 35, 45, 55, 65, 75, 85, 95 cm) for both profiles. For the TU-UP profile, the first sample was taken at 10 cm. We determined water retention at high matric potentials (0, $-3$, $-6$, $-10$, $-24$, $-46$ kPa) on the undisturbed samples by the multistep apparatus (van Dam et al., 1994). Afterward, these samples were dried and weighed to calculate the bulk density (BD; g cm$^{-3}$). We determined water retention at lower matric potentials ($-100$, $-300$, and $-1500$ kPa) on saturated disturbed samples by the Eijkelkamp pressure membrane apparatus (Klute, 1986). A total of 36 undisturbed and 18 disturbed samples were measured at the Hydro-physics Labora-

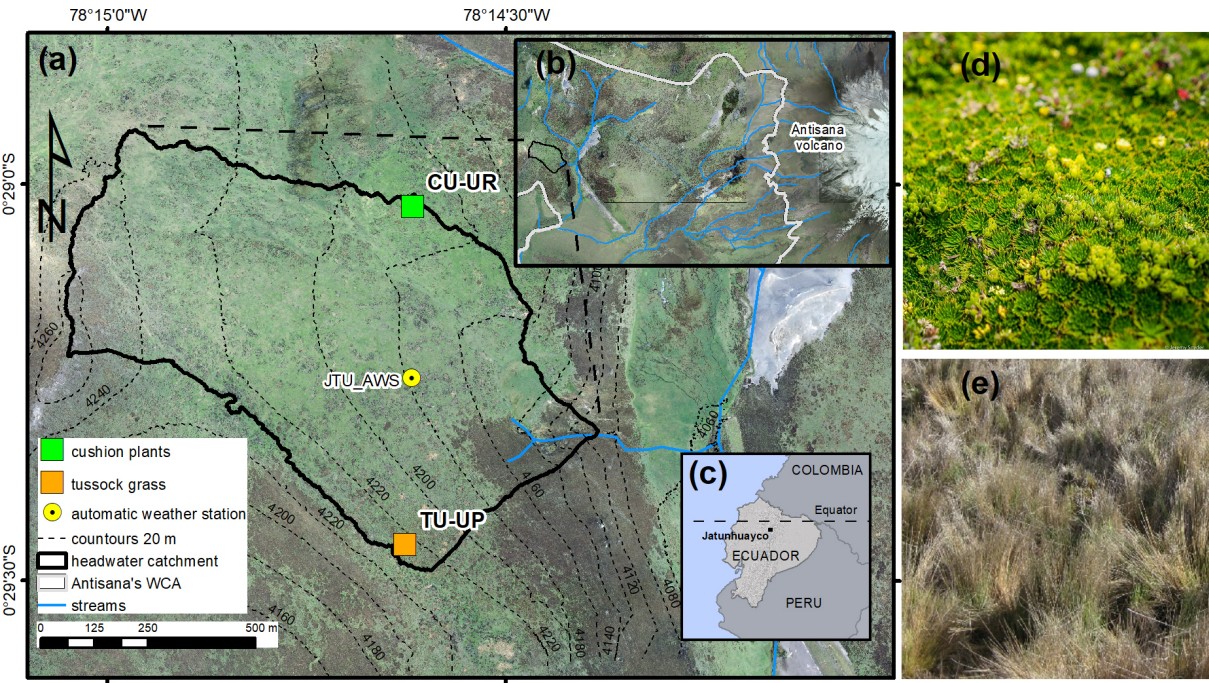

**Figure 1. (a)** Soil profiles CU-UR and TU-UP within the Jatunhuayco catchment, **(b)** Antisana's Water Conservation Area (WCA), **(c)** locations of study sites in Ecuador, **(d)** cushion-forming plants and **(e)** tussock grass. (Orthophotos from SIGTIERRAS, Ministerio de Agricultura y Ganadería, Ecuador; cushion-forming plant photo © Jeremy Snyder.)

**Table 1.** Description of soil profiles.

| Soil profile | CU-UR | TU-UP |
|---|---|---|
| Coordinates[a] | 0°29′1.69″ S, 78°14′37.69″ W | 0°29′27.94″ S, 78°14′37.07″ W |
| Altitude (m a.s.l.) | 4196 | 4225 |
| Mean slope[b] (%) | 2.0 | 6.5 |
| Soil horizon depth (cm) | | |
| A | 8–30 | 5–30 |
| 2A | 30–55 | 30–60 |
| 2BC | 55–80 | 60–85 |
| 3BC | 80–102 | 85–110 |
| Suction cups and soil water reflectometer installation depth (cm) | 20 40 80 | 25/20[d] 50/45[d] 100/95[d] |
| Surface coverage[c] (%) | | |
| Bare soil | $1.5 \pm 1.0$ | $3.3 \pm 4.2$ |
| Litter | $4.0 \pm 0$ | $33.7 \pm 21.4$ |
| Vegetation | $94.5 \pm 1$ | $63.0 \pm 21.3$ |
| Dominant vegetation species[c] (%) | $78.4 \pm 6.9$ *Azorella pedunculata* | $55.8 \pm 21.6$ *Calamagrostis intermedia* |

[a] Coordinates are shown as WGS84. [b] Mean value, based on three clinometer measurements (Eyeskey Optical Instrument Co., Ltd, China). [c] Mean value ± 1 standard deviation. [d] Installation depth of the suction cup and soil water reflectometer, respectively.

tory, University of Cuenca. Water retention is reported as the volumetric water content ($\theta$, cm³ cm⁻³) for a given matric potential $\psi_m$ (kPa) and was determined for saturation ($\theta_{SAT}$) at 0 kPa, field capacity ($\theta_{FC}$) at $-10$ kPa, and wilting point ($\theta_{WP}$) at $-1500$ kPa. The total available water ($\theta_{TAW}$) is calculated as the difference between water retention at field capacity and wilting point. The saturated hydraulic conductivity ($K_{SAT}$) was measured in replicates per soil horizon by the Guelph Permeameter (Soilmoisture Equipment Corp) using the two-head method (5 and 10 cm) (Reynolds and Elrick, 1985). The replicates were taken at >3 m from each other (Table S1 in the Supplement). A total of 17 $K_{SAT}$ measurements were taken at both soil profiles. Soil texture was determined by the laser-diffraction particle-sizing analyzer (LS 13 320, Beckman Coulter) at the Geo-Institute of KU Leuven. Sample preparation consisted in grinding and sieving (2 mm) the dry soil samples and removal of solutes and gypsum (if any) with demineralized water, carbonates with 10 % HCl, and organic matter with 35 % hydrogen peroxide. The soils were then treated with ultrasonics to disperse clays. A total of 10 disturbed samples corresponding to 1 sample per soil horizon were used for the texture analysis. The texture is expressed as a percent of the bulk soil (%) and is classified based on the following particle size ranges: sand (2000–50 µm), silt (50–2 µm), and clay (<2 µm).

Soil chemical properties were determined at the MOCA platform, Earth and Life Institute, UCLouvain, based on 10 disturbed samples taken at the mid-height of each hori-

zon for both profiles. The soil organic carbon (SOC) was measured on 5 d air-dried soil samples, which were sieved through a 2 mm mesh and crushed using a vibratory disk mill (Retsch RS200). SOC was measured by dry combustion with an Elementar Variomax elemental analyzer ($<0.1\%$ precision) and is expressed in percentage of dry weight soil. Soil pH was measured in a $1:5$ soil-to-water ($w/v$) suspension with a glass electrode. Cation exchange capacity (CEC, $\mathrm{cmol_c\,kg^{-1}}$) was measured by the ammonium acetate method (Pansu and Gautheyrou, 2006). Plant root abundance and diameters were characterized in the field per genetic horizons following the procedures of the World Reference Base for Soil Resources (IUSS Working Group WRB, 2014).

## 3.3 Hydrometry and water fluxes

An automatic weather station (JTU_AWS) recorded rainfall, air temperature and relative humidity, incoming solar radiation, and wind speed. To exclude potential field-scale variability in meteorology, we also measured at the TU-UP and CU-UR locations rainfall, air temperature, and relative humidity. Soil moisture, expressed as the volumetric water content (VWC, $\mathrm{cm^3\,cm^{-3}}$), was measured at three depths in each profile (Table 1) using water content reflectometers. Reflectometers were installed on the upslope-facing wall of the soil pits in a horizontal direction (Fig. 2b, d). They were calibrated for each horizon and vegetation type (for $\pm1\%$ accuracy, Sect. S2) following earlier work in páramo soils (Iñiguez et al., 2016; Ochoa-Sánchez et al., 2018). All variables were recorded at 5 min intervals and were further aggregated to daily values.

Water fluxes were measured in the 2A horizon of each soil profile using suction (with wick, FXW) and non-suction (without wick, FX) fluxmeters (Fig. 2a, d). Although wick lysimeters have been extensively used to measure water fluxes (Singh et al., 2018), they may overestimate fluxes when compared to non-suction lysimeters (van der Velde et al., 2005; Weihermüller et al., 2007). The design of both fluxmeters was based on Gee et al. (2009) and van der Velde et al. (2005). They consisted of polyvinyl chloride (PVC) tubes of 20 cm diameter inserted into the soil to a depth of 50 cm. Their lower end is in contact with a funnel filled with soil and covered with a nylon mesh of 20 µm at the bottom. For the FXW fluxmeter, one round fiberglass wick (1381 Pepperell Braiding Company) was unbraided and placed below the nylon mesh of the funnel and extended to a total length of 50 cm. The lowest part of the funnel was connected to a 1-gallon CE1 plastic container to collect water draining from the soil. The water volume was recorded biweekly with a resolution of 1 mL and converted into flux by dividing it by the internal cross-sectional area of the PVC tube (283.5 $\mathrm{cm^2}$).

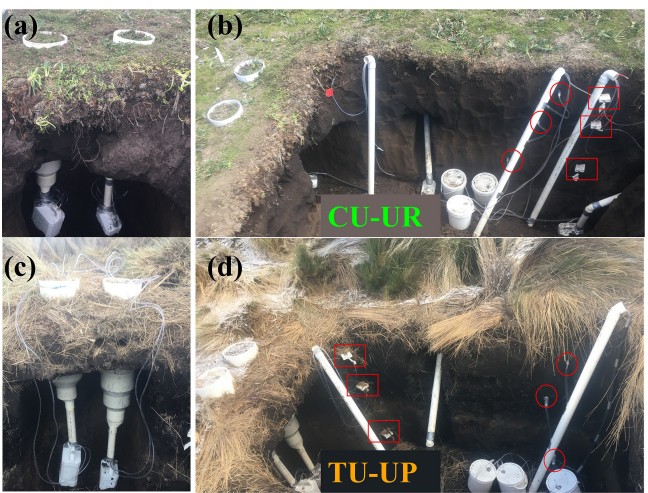

**Figure 2.** Photos of the soil pits under cushion-forming plants (CU-UR, **a**, **b**) and tussock grasses (TU-UP, **c**, **d**) showing the fluxmeters ((**a**), (**c**)); water content reflectometers (red rectangles) and suction cups (red circles) ((**b**), (**d**)).

## 3.4 Water flux modeling

The simulation of soil hydrology at the profiles was conducted using the one-dimensional HYDRUS-1D model version 4.17 (Šimůnek et al., 2018) for the period between December 2018 and March 2021. We used the bimodal van Genuchten model (Durner, 1994) to represent the water retention and hydraulic conductivity functions (Priesack and Durner, 2006; Dettmann et al., 2014) of the soil matrix. The upper-boundary condition was set to an atmospheric boundary with surface runoff and the lower boundary to free drainage. The model was set to run at a daily timestep with calibration (22 December 2018 to 5 March 2020) and validation (6 March 2020 to 12 March 2021) periods. The initial soil hydraulic parameters were defined from field measurements and/or previously fitted bimodal van Genuchten models (Páez-Bimos et al., 2022). To reduce the number of soil hydraulic parameters, we performed a global sensitivity analysis using the variance-based Sobol method (Sobol, 2001). The sensitivity analysis used parameter ranges defined from the field measurements ($\theta_S$, $\theta_r$, $K_{\mathrm{SAT}}$; Páez-Bimos et al., 2022) or from the literature ($\alpha$, $n$, $w_2$, $\alpha_2$, $n_2$; Dettmann et al., 2014) for bimodal porous degraded organic soils. The tortuosity parameter ($\tau$) was set to 0.5, as proposed by Mualem (1976) and recommended for soils with SOC $< 18\%$ (Dettmann et al., 2014). Table S4 contains the details on the sensitive parameters and the parameter ranges. Using the sensitive parameters along with the initial parameters, we carried out an inverse numerical modeling for the calibration period using the Marquardt–Levenberg algorithm (Šimůnek and Hopmans, 2002) based on the observed and simulated soil VWC at three depths. Section S3 describes the model setup, sensitivity analysis, and calibration. The good-

ness of fit was assessed by the coefficient of determination ($R^2$), root mean square error (RMSE), and Kling–Gupta efficiency (KGE). A KGE value greater than $-0.41$ indicates that the model predicts better than the mean of the observations, while a value of 1 indicates a perfect agreement between observed and simulated values (Knoben et al., 2019). Apart from the validation of the model simulations, an independent validation was performed using the field-based water flux measurements at the 2A horizon from the FXW and FX fluxmeters. These values were collected over 24 field visits during the period April 2019 to March 2020 and compared to the simulated water flow at 50 cm depth that was aggregated at the same time interval (Table S3). Simulated and measured water fluxes were analyzed by Spearman correlation since most water flux variables did not show normality.

The uncertainty in the modeled water fluxes was assessed using the generalized likelihood uncertainty estimation (GLUE) method (Beven and Binley, 1992). First, we selected the sensitive parameters, defined by the Sobol method, to randomly generate 50 000 parameter sets for each model (Kettridge et al., 2015; Selle et al., 2011). Second, we generated the sensitive parameters in the range of the fitted values plus or minus 2 standard deviations, as reported from the inverse modeling. We assumed uniform distributions for all parameter sets. Third, we ran the models for the 50 000 parameter sets for the calibration and validation periods. We compared the simulated and observed soil moisture using the $R^2$ and KGE. We determined the behavioral parameter sets, discarding simulations where $R^2 < 0.2$ and KGE $< 0$ (Houska et al., 2014). All model simulations, sensitivity analyses, uncertainties, and calibrations were carried out in the R programming language (R Development Core Team, 2010) by adapting the R packages sensitivity and hydrusR (Acharya, 2020; Pujol et al., 2017).

To examine the influence of vegetation on the water partitioning, we established a soil water balance ($\Delta S = P - \text{ETa} - D$) for the upper 1 m of the soil profiles and derived the following soil water balance components at daily and annual timescales: the rainfall ($P$), the actual evapotranspiration (ETa), the soil water storage ($\Delta S$), and the deep drainage ($D$). ETa was derived from potential evapotranspiration (ETp) according to the pressure head at the soil surface and soil moisture. We calculated ETp based on the Penman–Monteith equation, as implemented in HYDRUS-1D (Šimůnek et al., 2018), using daily meteorological data from station JTU_AWS: incoming solar radiation, wind speed, relative humidity, and minimum and maximum air temperature. We left default values of meteorological parameters for cloudiness and emissivity for longwave radiation and Ångström values for shortwave radiation. We used 12 h for daily sunshine and did not consider data for crops. The albedo was set to 0.14, which is the average of the albedo values that were reported earlier for the Ecuadorian páramo (0.11–0.17; Montenegro-Díaz et al., 2022; Minaya et al., 2018).

## 3.5 Characterization of soil solutions

### 3.5.1 Soil solution collection and pretreatment

Soil porewater was sampled with suction cup samplers at three depths per profile as detailed in Table 1. The samplers are 0.50 m in length, with a porous ceramic cup of a maximum 1 µm pore size. We installed the suction lysimeters subhorizontally (with a 5 % downward inclination) on the upslope-facing wall of the soil pit (Fig. 2b, d). Soil solutions were collected in 500 mL glass bottles wrapped in aluminum paper, closed with rubber stoppers, and placed inside plastic containers. All tubing was shielded from sunlight. Suction cup samplers were installed between November and December 2018, and soil porewater samples were analyzed over the period from April 2019 to March 2020. This left ample time for the porous cups to equilibrate with the soil conditions (Dere et al., 2019). Soil porewater was collected biweekly, and a vacuum of 45–50 kPa was applied after every collection. Soil solutions were filtered through 0.45 µm mixed cellulose ester filters of 47 mm (S-PAK® Membrane Filter). The material was prewashed by passing $> 150$ mL of ultrapure water through the filter to avoid leachate of dissolved organic carbon from the cellulose (Khan and Subramania-Pillai, 2007). Filtered samples were split into two 30 mL plastic bottles, which were previously washed with a 0.5 M $HNO_3$ solution and rinsed with ultrapure water three times before use. One split was left unpreserved for analyses of anions and dissolved organic carbon (DOC), and the other split was acidified to pH $< 2$ by adding three drops of concentrated (68 %) nitric acid for cation analysis. All samples were refrigerated at $\sim 4$ °C during storage.

### 3.5.2 Water chemical analyses

In the field, the electric conductivity, water temperature, and pH of all the samples were measured. The alkalinity was determined in a field laboratory setup at 4010 m a.s.l. on the same day of collection using the ASTM D1067-16 (2016). For a selection of 13 sample collections (of a total of 24), the cation, anion, and dissolved organic carbon were determined at the MOCA platform, Earth and Life Institute, UCLouvain (Table S3). Cation concentrations (total Ca, Mg, Na, K, Al, Fe, and dissolved silica – DSi) were determined by inductively coupled plasma optical spectroscopy (ICP-OES, Thermo Fisher iCAP 6000 series, limit detection $< 1$ ppb). DOC concentrations were determined by the combustion catalytic oxidation/NIR CE2 method (TOC-L series, SHIMADZU) with a detection limit of 4 ppb. Anion concentrations ($Cl^-$, $NO_3^-$, $SO_4^{2-}$) were determined by ion chromatography (Dionex ICS 2000, Thermo Fisher) with a detection limit of 200 ppb. All solute concentrations are reported (mg $L^{-1}$). We removed outliers by eliminating concentrations larger (smaller) than 2 standard deviations above (below) the average (Dere et al., 2019). The charge balance

error (CBE) was calculated for each available sampling time. Only 2 soil solutions out of 66 with more than two ion concentrations equal to NA were discarded for CBE calculation.

Daily water fluxes simulated by the model were aggregated into biweekly intervals by addition. Biweekly solute fluxes were calculated by multiplying the solute concentrations by the biweekly water fluxes at the corresponding depth and time interval. To obtain annual solute fluxes, biweekly solute fluxes at all sampling times ($n = 24$) were aggregated over the period 29 March 2019 to 4 March 2020. For sampling times with missing solute concentrations ($n = 11$), we interpolated these concentrations based on linear regressions between biweekly water fluxes and the available solute concentrations (Fig. S1). Contemporary weathering fluxes are the sum of major cations ($Ca^{2+}$, $Mg^{2+}$, $Na^+$, $K^+$) and DSi fluxes ("T.Cat.").

To propagate the uncertainty of the water fluxes to the solute fluxes, we selected randomly 10 000 behavioral model runs (from the total of 50 000 runs) of the biweekly water fluxes and used them in a linear regression model together with the available biweekly solute concentrations to complete the missing biweekly solute concentrations. We included the uncertainty of the linear regression by generating 100 random fitting parameters for each selected behavioral run. We determined $1 \times 10^6$ biweekly solute fluxes by multiplying the water fluxes by the solute concentrations. Finally, we aggregated the biweekly solute fluxes to annual values and report the mean annual solute fluxes along with the 95 % confidence intervals (mean $\pm 2$ standard deviations).

## 3.6 Statistical analysis

The Mann–Whitney $U$ test was applied to test for significant differences ($p < 0.05$) in solute concentrations and solute fluxes between two groups (vegetation types: cushion-forming plants vs. tussock grasses). This test was applied for the A and 2A horizons, as these horizons correspond to the parts of the soil profiles where the root systems are mostly developed. The Mann–Whitney $U$ test is suited for small non-normal distributed datasets, as evidenced by the Shapiro–Wilk test ($p < 0.05$). To analyze the effects of seasonality on soil water balance and weathering, we separated dry, intermediate and wet periods based on rainfall data. The dry period was defined as the months with rainfall values less than the 25th percentile of the annual average rainfall (January, July–September 2019 and March and October 2020). The wet period was assumed to correspond to the months when the monthly rainfall exceeded the 75th percentile CE3 of the annual average rainfall (April–May and October–November 2019, November–December 2020 and February 2021). The remaining months were defined as being "intermediate" (Fig. S2). All statistical analyses and plots were carried out in the R programming language (R Development Core Team, 2010).

# 4 Results

## 4.1 Depth variation of soil hydrophysical and chemical properties

There are clear differences in the properties of the A and 2A horizons between the CU-UP and TU-UP soils. The depth variation of the mean total available water ($\theta_{TAW}$) is different for the two vegetation types, with a maximum $\theta_{TAW}$ of 0.50 and 0.52 $cm^3 cm^{-3}$ for the A horizon of CU-UR and the 2A horizon of TU-UP, respectively (Fig. 3b). The depth variation of $\theta_{TAW}$ is inverse to the BD (Fig. 3c). The mean $K_{SAT}$ is highest under cushion plants: the elevated values of 175 $mm h^{-1}$ measured for the A horizon (Fig. 3a) are in contrast with the 1 order of magnitude lower value in the 2A horizon and 2 orders of magnitude lower in the 2BC and 3BC horizons (15.6 and 1 $mm h^{-1}$, respectively). Under tussock grass, the depth variation of $K_{SAT}$ is more uniform, with values between 5 and 12 $mm h^{-1}$ for the A, 2A, and 2BC horizons and a decrease to 1 $mm h^{-1}$ for the 3BC horizon. Mean $K_{SAT}$ depth variation resembles the vertical distribution of root diameter and abundance (Fig. 3e, f). Water retention at saturation and field capacity showed a similar depth variation to $\theta_{TAW}$, whereas water retention at wilting point showed negligible differences by vegetation type (Fig. S3, Table S1).

The soil BD in the upper three soil horizons (A to 2BC) is different by vegetation type (Fig. 3c). Under cushion plants, the lowest value of 0.43 $g cm^{-3}$ is measured in the topsoil, and values then systematically increase with depth to 75 cm. Under tussock grass, there is more local depth variation than under cushions: the lowest value (0.62 $g cm^{-3}$) is measured in the 2A horizon, but local maxima occur at 35 and 85 cm depth. The maximum BD ($\sim 1.25 g cm^{-3}$) of both profiles is situated at the 2BC–3BC interface (70–90 cm depth). The soil texture of the three upper horizons is similar under both vegetation types: the sand and silt fractions are dominant (35 % to 63 %), while the clay content is less than 8 %. In the 3BC horizon, a higher silt and clay content is observed under tussock grasses (Fig. S3, Table S2).

In the topsoil, SOC and CEC are highest and decrease with depth. In the A horizon, SOC and CEC are higher under cushion plants. Under cushion plants, SOC decreases strongly from 9 % to 2 %, whereas below tussocks, it decreases from 6 % to 3 % with a local minimum (1 %) at the 2BC horizon (Fig. 3d, Table S2). Soil pH is acidic and lower (4.6 to 5) under cushion plants compared to tussock grass in the entire profile (Fig. 3). Also, the depth variation differs: under cushion plants, the pH decreases from the topsoil to the 2A horizon and then increases to lower depths, whereas under tussock grasses there was a consistently increasing pH with depth (Fig. 3g, Table S2). In the A horizon, CEC is highest (35 $cmol_c kg^{-1}$) under cushion plants, compared to 26 $cmol_c kg^{-1}$ under the tussock profile. The highest cation exchange capacity at the upper horizon decreased with depth until the 2BC horizon, and then it slightly increased at the

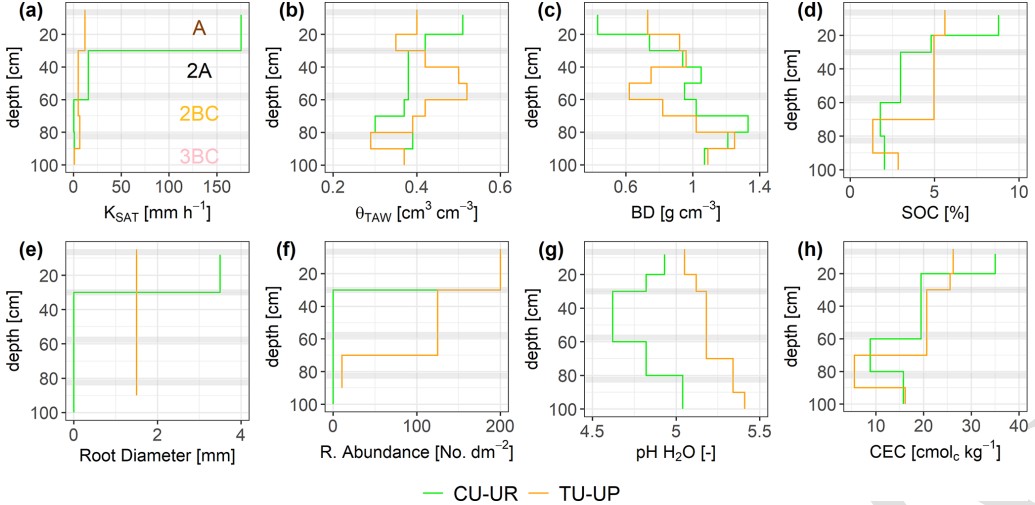

**Figure 3.** Depth-specific distribution of soil properties: **(a)** saturated hydraulic conductivity ($K_{SAT}$), **(b)** total available water ($\theta_{TAW}$), **(c)** dry bulk density (BD), **(d)** soil organic carbon (SOC), **(e)** root diameter, **(f)** root abundance, **(g)** soil pH, and **(h)** cation exchange capacity (CEC). Horizontal gray bars indicate the boundaries between soil horizons. The information of the soil pits under cushion-forming plants (CU-UR) is plotted in green color, while the information from soils under tussock grasses (TU-UP) is plotted in orange.

3BC horizon for both soil profiles. Details on soil properties are included in the Supplement (Table S2). We have observed the same vertical distribution of soil properties ($K_{SAT}$, $\theta_S$, $\theta_{FC}$, $\theta_{WP}$, $\theta_{TAW}$, BD, and SOC) and root characteristics in six other soil profiles of the same subcatchment in Jatunhuayco (Páez-Bimos et al., 2022). Based on these observations, we assume that the differences that were observed between the two profiles (CU-UR and TU-UP) are indicative of the differences between soil profiles under cushion-forming plants and tussock grasses in similar topographic positions.

### 4.2 Model simulations and independent validation

Sensitive parameters were identified from the initial 72 parameters (for each soil profile) by the Sobol method, whereby parameter values varied by vegetation type (full details are given in Sect. S3). For the soil profile under cushion-forming plants, the most sensitive soil hydraulic parameters are the $n$ parameters along depth, whereas for the tussock profile the most sensitive parameters are related to the soil hydraulic properties of the upper soil horizon (10–25 cm especially) (Table 2). The calibrated soil hydraulic parameters obtained from the inverse modeling are shown in Table 2. The standard deviations for most parameters are small, indicating that the inverse modeling approach gave stable parameter estimates.

After calibration of the Hydrus-1D model with the bimodal van Genuchten approach, based on observed and simulated soil volumetric water contents, the model performance for the calibration period resulted in $R^2$ values of 0.63 to 0.90, RMSE $\leq 0.02$ cm$^3$ cm$^{-3}$, and KGE values of 0.39–0.86 for the A and 2A horizons and 0.07–0.12 for the 2BC and 3BC horizons (Fig. 4). For the validation period, the model performance is slightly higher than for the calibration ($R^2$:

0.49–0.91, RMSE $\leq 0.02$ cm$^3$ cm$^{-3}$, KGE: 0.35–89 for the A and 2A horizons and 0.08–0.71 for the 2BC and 3BC horizons), indicating that the soil hydrological processes are rather well represented (Fig. 4). Despite the fact that the KGE values are generally low, they are above $-0.41$, indicating that the model adequately predicts the mean of the observations (Knoben et al., 2019). Figure 4 includes the range of uncertainty in the modeled volumetric water contents based on the 44 150 to 50 000 behavioral model runs for each soil horizon and illustrates that the model performance decreases in the lower horizons. For the cushion plant profile, the fitted soil hydraulic parameters are related to the slope of the water retention function ($n$) along with soil profile depth, whereas for the tussock grass profile most fitted parameters are related to the soil hydraulic properties in the A horizon (Table 2). The standard errors of the fitted hydraulic parameters were 1 order of magnitude lower than their mean value for most cases (except for parameter $n$ at 35, 55, and 65 cm depth at CU-UR). This indicates that the inverse approach gave, in general, stable estimates (Table S6). The mass balance error in the numerical solution of the model was lower than 1 % for both profiles.

Soil moisture (VWC) varied in depth by vegetation type (Fig. 4, Table 4). Mean annual soil moisture is highest in the A horizon (0.63 cm$^3$ cm$^{-3}$) and declines strongly with depth to the 2BC horizon (0.40 cm$^3$ cm$^{-3}$) below cushion plants, whereas it is highest in the 2A horizon (0.63 cm$^3$ cm$^{-3}$) and decreases slightly to the upper (0.59 cm$^3$ cm$^{-3}$) and lower (0.55 cm$^3$ cm$^{-3}$) horizons below tussock grasses. Soil moisture response to rainfall events and dry periods is more dynamic under cushion vegetation throughout the soil profile (Fig. 4a).

**Table 2.** Fitted soil hydraulic parameters. The optimal model fit is given with 1 standard deviation (SD).

| CU-UR | | | TU-UP | | |
|---|---|---|---|---|---|
| Parameter | Depth (cm) | Fitted value (1 SD) | Parameter | Depth (cm) | Fitted value (1 SD) |
| $n$ (–) | 15 | 2.50 (0.09) | $\alpha$ (1 cm$^{-1}$) | 10 | 0.028 (0.0002) |
| $n$ (–) | 25 | 1.21 (0.005) | $n$ (–) | 10 | 2.50 (0.04) |
| $n$ (–) | 35 | 2.50 (0.51) | $K_{SAT}$ (cm d$^{-1}$) TS3 | 10 | 4.96 (0.20) |
| $n$ (–) | 45 | 1.23 (0.003) | $w_2$ (–) | 10 | 0.001 (0.0007) |
| $n$ (–) | 55 | 2.50 (0.54) | $\alpha_2$ (1 cm$^{-1}$) | 10 | 0.007 (0.0008) |
| $n$ (–) | 65 | 2.50 (0.61) | $n_2$ (–) | 10 | 1.50 (0.19) |
| $n$ (–) | 85 | 1.26 (0.003) | $\alpha$ (1 cm$^{-1}$) | 25 | 0.003 (0.00004) |
| | | | $n$ (–) | 25 | 2.23 (0.09) |
| | | | $\alpha$ (1 cm$^{-1}$) | 65 | 0.018 (0.0007) |
| | | | $n$ (–) | 65 | 2.45 (0.10) |
| | | | $\alpha$ (1 cm$^{-1}$) | 75 | 0.004 (0.0003) |
| | | | $n$ (–) | 75 | 2.50 (0.22) |

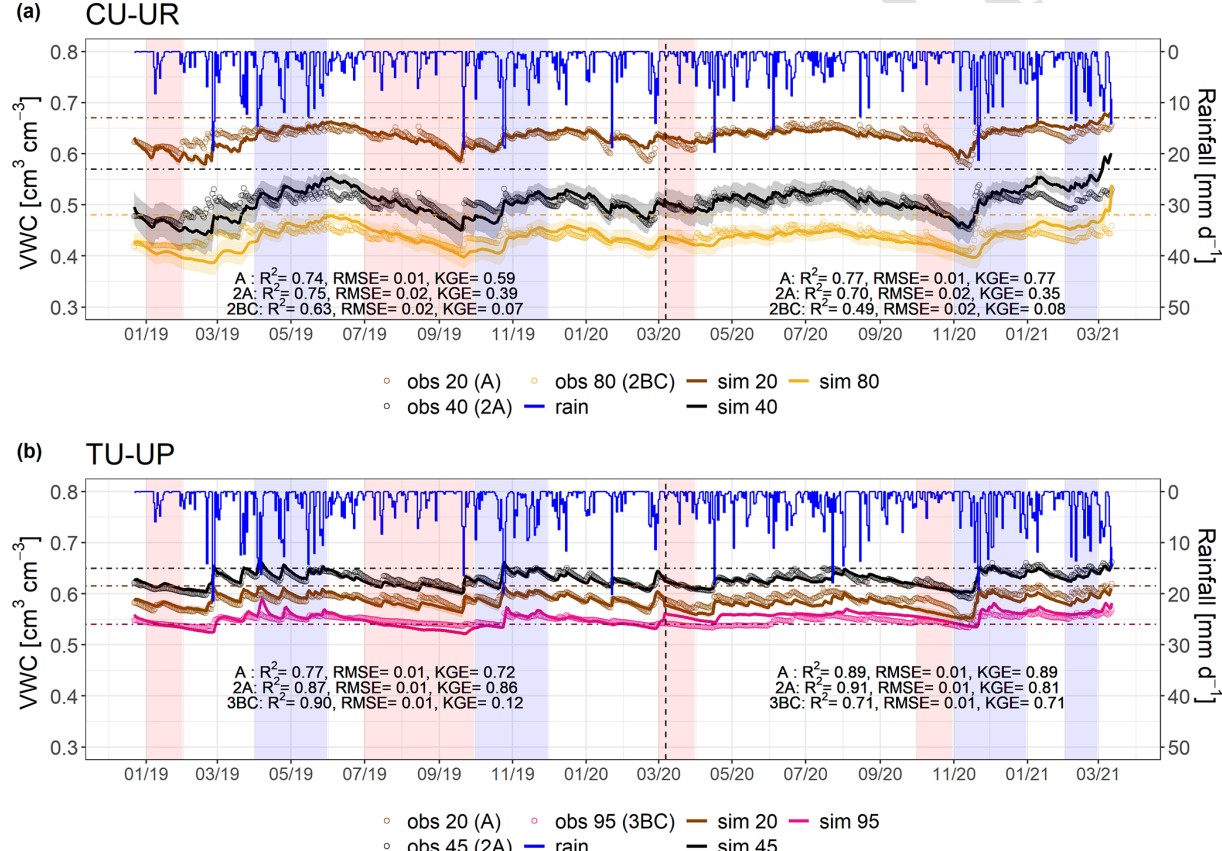

**Figure 4.** Simulated (thin line) and observed (dots) volumetric water content (VWC) for **(a)** the CU-UR profile and **(b)** the TU-UP profile. The rainfall is represented by the blue line. Brown and black colors represent the A (at 20 cm) and 2A (at 40/45 cm) horizons for both soil profiles. The yellow and pink lines represent the 2BC (at 80 cm) and 3BC (at 95 cm) horizons for the CU-UR and TU-UP profiles, respectively. The vertical dashed line separates the calibration and validation periods. Horizontal dot-dashed lines indicate field capacity ($\theta_{FC} = -10$ kPa) at the corresponding depth. Red stripes represent months defined as dry periods, while blue stripes represent months defined as wet periods. Colored lines show the optimal model VWC estimate $+ 95\%$ CI.

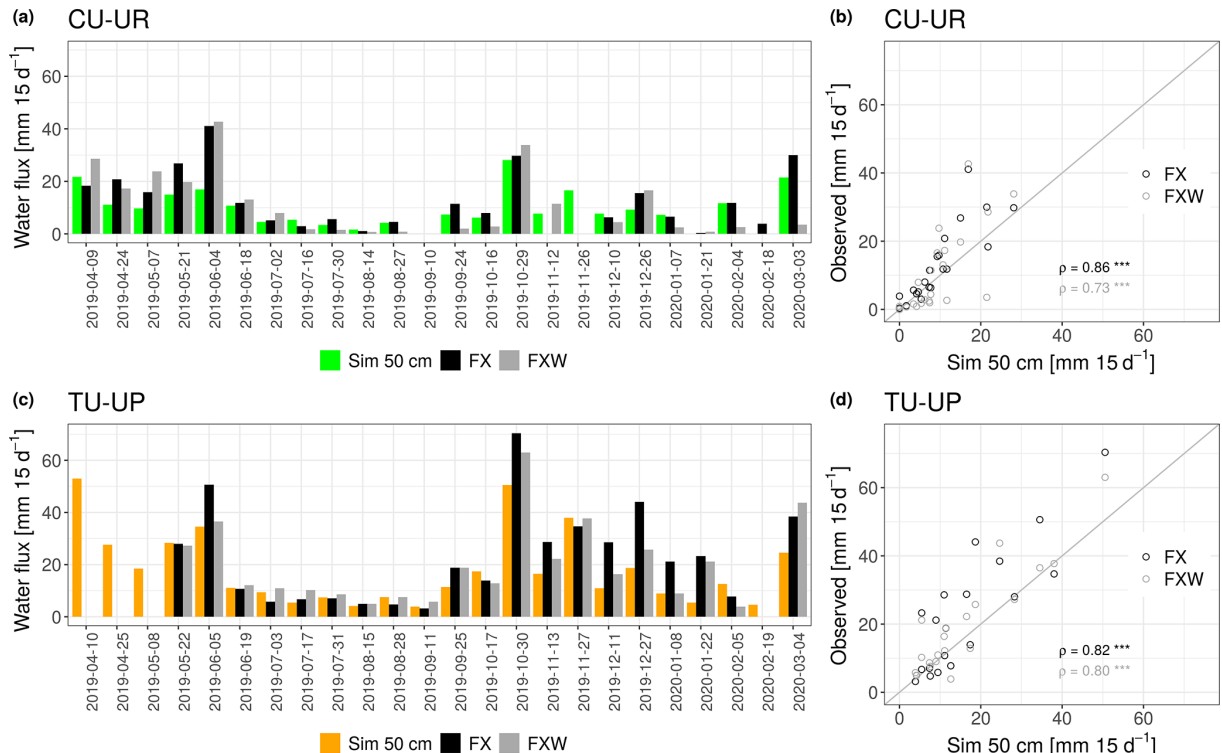

**Figure 5.** Panels **(a)** and **(c)** show the biweekly temporal series of simulated (Sim) and observed water fluxes (FX, FXW). Panels **(b)** and **(d)** show the correspondence between the simulated and observed water fluxes. Water fluxes were measured by two fluxmeters (FX: without wick, FXW: with wick) installed at 50 cm depth in each profile. The Spearman correlation coefficients ($\rho$) and their significance are given in the scatterplots (with levels of significance *, **, and *** corresponding to $p<0.05$, $p<0.01$, and $p<0.001$).

Besides the internal calibration and validation with daily volumetric water contents, the model performance was also evaluated with biweekly independent observations obtained from the two water fluxmeters installed at 50 cm depth (2A horizon) of each profile (Fig. 5). Simulated and observed water fluxes under tussock grass are systematically higher than fluxes under cushion plants over the study period (Fig. 5a, c). There is a significant correlation between the simulated and observed water fluxes in the 2A horizon, as all Spearman correlation coefficients are $>0.73$ with $p<0.001$ (Fig. 5b, d). Nevertheless, the comparison also shows that the simulations tend to underestimate water fluxes at higher rates ($>20\,\mathrm{mm}\,15\mathrm{d}^{-1}$).

### 4.3 Soil water dynamics under different vegetation covers

The soil water dynamics of both profiles were compared based on the time series (December 2018–March 2021) of rainfall ($P$), ETa, drainage ($D$), and soil water storage ($S$) at a daily scale (Fig. 6). Rainfall was evenly distributed over the study period, and the mean annual values (2019–2020) did not differ between the two sites ($687\pm11$ for CU-UR and $726\pm68$ mm for TU-UP). The daily rainfall intensity reached up to $21\,\mathrm{mm}\,\mathrm{d}^{-1}$. There exist clear differences in mean an-

nual water partitioning and soil water balance between soils under cushion-forming plants and tussock grasses (Table 3). Mean annual ETa is higher for cushion plants ($522\pm75$ mm) than for tussock grass ($308\pm65$ mm). The difference is highest during the dry periods (January 2019; July–September 2019; March 2020; October 2020), when the ETa under cushion plants is 2 times higher than under tussock grasses (Fig. 6c, Table S7).

Deep drainage ($D$) is – on an annual average – 3 times higher under tussock grasses ($405\pm183$ mm) than under cushion plants ($127\pm13$ mm). The largest differences occur during the rainy periods (April–May 2019; October–November 2019; November–December 2020; February 2021), when deep drainage under tussock grasses can be up to 13-fold higher than under cushions (Table S7, Fig. 6d). In contrast to the tussock grass, the deep drainage under cushion-forming plants is steady over time and less reactive to rainfall inputs. Furthermore, the soil water storage in the soil profile ($S$) is more dynamic under cushion plants than under tussock grasses (Fig. 6e), showing a larger range for the cushion plant profile ($34\pm43$ mm) compared to tussock grass ($9\pm30$ mm).

The mean annual water fluxes below tussock grasses are higher than under cushion plants for all horizons (Table 4). Under cushion plants, the mean annual water flux is highest

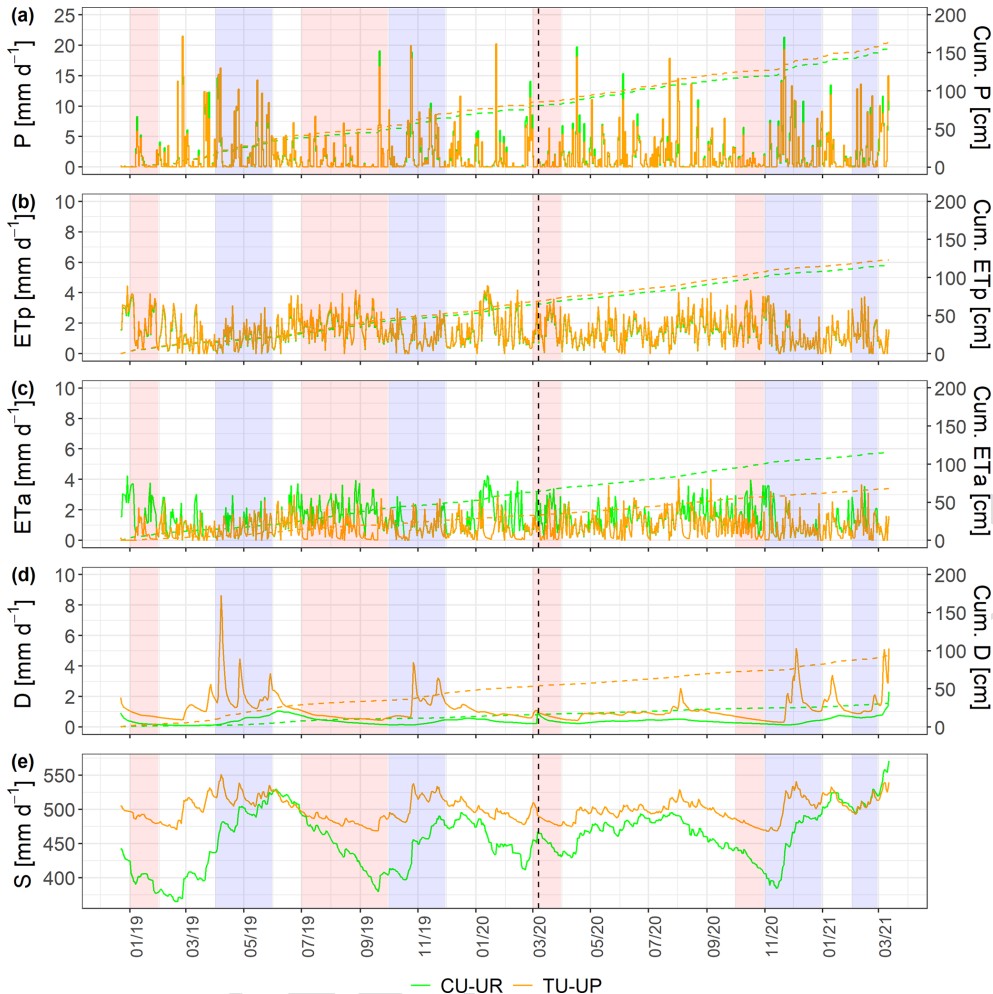

**Figure 6.** Soil water balance at a daily scale: **(a)** rainfall ($P$), **(b)** potential evapotranspiration (ETp), **(c)** actual evapotranspiration (ETa), **(d)** deep drainage ($D$), and **(e)** soil water storage ($S$) (vertical dashed line separates calibration and validation periods). The green and brown lines represent the CU-UR and TU-UP profiles, respectively. Red stripes represent months defined as dry periods, while blue stripes represent months defined as wet periods.

**Table 3.** Mean annual water balance (January–December 2019 to January–December 2020) and the corresponding percentage of rainfall input (mean $\pm 2$ standard deviations).

| Soil profile | $P$ (mm) | ETp (mm) | ETa (mm) | $D$ (mm) | $\Delta S$ (mm) | ETa (mm d$^{-1}$) |
|---|---|---|---|---|---|---|
| CU-UR | $686.6 \pm 11.0$ | $526.0 \pm 63.6$ ($76.6 \pm 12\%$) | $522.1 \pm 74.6$ ($76.0 \pm 14\%$) | $127.0 \pm 12.6$ ($18.5 \pm 9.9\%$) | $34.1 \pm 43.2$ ($5.0 \pm 126\%$) | $1.43 \pm 0.18$ |
| TU-UP | $726.3 \pm 68.4$ | $558.5 \pm 70.6$ ($77 \pm 13\%$) | $308.3 \pm 65.4$ ($42.5 \pm 21\%$) | $404.8 \pm 183.4$ ($55.7 \pm 45\%$) | $9.4 \pm 29.6$ ($1.3 \pm 314\%$) | $0.84 \pm 0.18$ |

in the A horizon ($357 \pm 44$ mm) and decreases with depth to the 3BC horizon (127 mm). In contrast, the mean annual water fluxes are more constant with depth under tussock grasses (420 to 405 mm). Under cushion plants, the A horizon is highly responsive to rainfall input (Fig. 7b), showing a high infiltration capacity; conversely, the lower horizons show an

attenuated response. In contrast, under tussock grass, water flux at the uppermost horizon is higher but less responsive to rainfall events and shows larger recessions once rainfall has stopped (Fig. 7d). Moreover, water fluxes below tussock grass show similar responses in all horizons in depth.

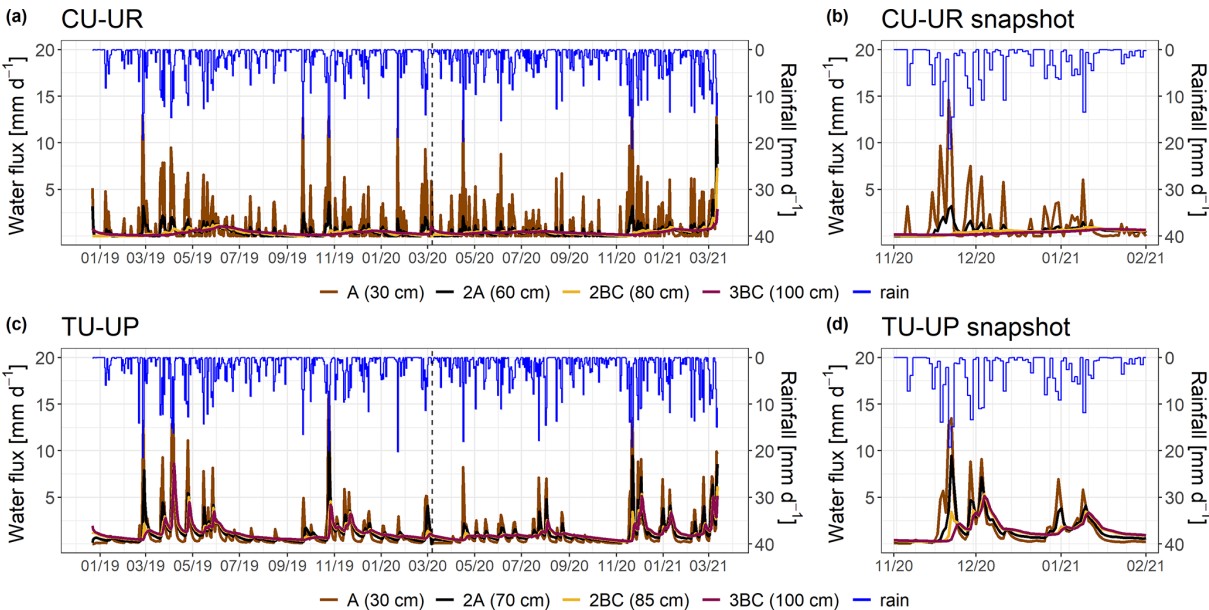

**Figure 7.** Simulated water fluxes over the period December 2018–March 2021 **(a, c)** for the two soil profiles. A snapshot is shown in panels **(b, d)** to highlight the response during rainfall events. The rainfall is represented by the blue line. Brown and black colors represent the A and 2A horizons for both soil profiles. The yellow and pink lines represent the 2BC and 3BC horizons for the CU-UR and TU-UP profiles, respectively. The water fluxes are derived from the bottom of the soil horizons.

**Table 4.** Mean annual simulated water fluxes and measured volumetric water content (January–December 2019 to January–December 2020) by soil horizon (mean $\pm 2$ standard deviations).

| Soil horizons | Water flux (mm) | Soil moisture ($cm^3\,cm^{-3}$) | Water flux (mm) | Soil moisture ($cm^3\,cm^{-3}$) |
|---|---|---|---|---|
| | CU-UR | | TU-UP | |
| A | $356.7 \pm 44.4$ | $0.63 \pm 0.00$ | $420.2 \pm 161$ | $0.59 \pm 0.00$ |
| 2A | $157.0 \pm 59.0$ | $0.50 \pm 0.00$ | $407.0 \pm 176$ | $0.63 \pm 0.00$ |
| 2BC | $124.5 \pm 29.8$ | $0.43 \pm 0.00$ | $404.8 \pm 182$ | – |
| 3BC | $127.0 \pm 12.6$ | – | $404.7 \pm 183$ | $0.55 \pm 0.00$ |

## 4.4 Solute concentrations and fluxes

The solute concentrations at both soil profiles generally decrease in the order $HCO_3^- > DSi > NO_3^- > DOC > Ca \approx SO_4^{2-} > Na > Mg \approx K \approx Cl^- \gg Al \approx Fe$ (Fig. 8, Table S8). The mean charge balance error was negative from $-5.9 \pm 6.8$ to $-11.9 \pm 7.3$ %, except at CU-UR in the A horizon ($12.9 \pm 7.7$ %). The mean DOC and Al concentrations are 1 order of magnitude higher ($p \leq 0.001$) in the A horizon below cushion plants (47 and $2\,mg\,L^{-1}$) than under tussock grass (3 and $0.1\,mg\,L^{-1}$). In the 2A horizon, the differences in mean DOC and Al concentrations ($p \leq 0.001$) are less pronounced, with concentrations under cushions (10 and $0.14\,mg\,L^{-1}$) being higher than under tussock grass (3 and $0.07\,mg\,L^{-1}$; Fig. 8f–g). While the DOC and Al concentrations further decrease in the soil water extracts of the 2BC horizon under cushion plants, they remain stable with depth under tussock grass (Table S8). In the A horizon, Ca and Mg concentrations are higher ($p \leq 0.001$) under tussock grass than under cushion plants; conversely, K concentration is higher for cushion plants. Below 30 cm, there are no significant differences by vegetation type (Fig. 8a, b, e).

Despite significant differences in $HCO_3^-$, Na, and DSi concentrations in the A and 2A horizons between the two vegetation types, they are not consistent with depth (Fig. 8c, d, h). DSi concentrations increase with depth for soils under both vegetation types. Under CU-UR, bicarbonate concentrations seem to follow the depth distribution of the roots, that is, higher $HCO_3^-$ values at the A horizon where roots are abundant and rapidly decreasing $HCO_3^-$ concentrations below the rooting zone. This trend is not clear for TU-UP. The Na concentration in the soil solutes varies only between

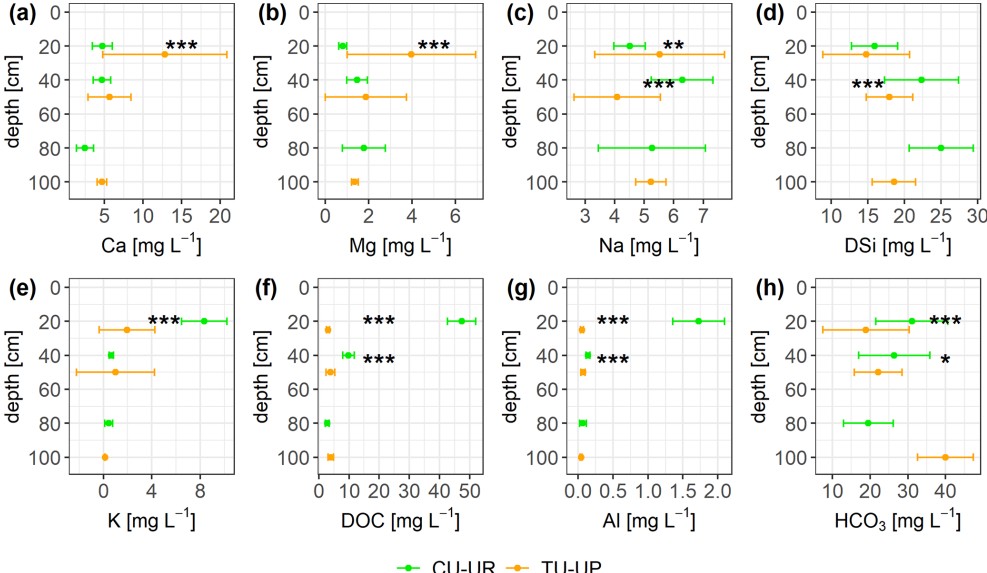

**Figure 8.** Solute concentrations (mean ±2 standard deviations) integrated over biweekly intervals for both soil profiles. The Mann–Whitney $U$ test was applied for differences between vegetation types, with levels of significance *, **, and *** corresponding to $p<0.05$, $p<0.01$, and $p<0.001$.

3 and $7\,\mathrm{mg\,L^{-1}}$ and exhibits no clear trends with depth and no consistent influence of vegetation.

The annual solute fluxes of cations and DSi (T.Cat) are taken here as a proxy for contemporary soil chemical weathering rates following Clow and Drever (1996), Guicharnaud and Paton (2006), and White et al. (2005). The annual fluxes below the TU-UP profile ($14–17\,\mathrm{g\,m^{-2}\,yr^{-1}}$) are systematically higher than under CU-UR ($5–14\,\mathrm{g\,m^{-2}\,yr^{-1}}$; Table 5). The dominant components are $HCO_3^-$ ($3–20\,\mathrm{g\,m^{-2}\,yr^{-1}}$) and DSi ($4–9\,\mathrm{g\,m^{-2}\,yr^{-1}}$). Remarkably, the solute flux in the A horizon below the CU-UR profile is principally composed of DOC with an annual flux of $20\,\mathrm{g\,m^{-2}\,yr^{-1}}$. Under CU-UR, the annual solute fluxes decrease strongly with depth, while they remain constant with depth under TU-UP (Fig. S5). This depth variation of annual solute fluxes resembles the trends of mean annual water fluxes (Table 4).

Most solute fluxes differ by vegetation type in the uppermost A horizon, with exception of the Na and DSi fluxes (Fig. 9, Table S9). At the A horizon under the CU-UR profile, the mean biweekly solute fluxes of DOC and total Al are 1 order of magnitude higher ($797\pm644$, $28\pm20\,\mathrm{mg\,m^{-2}\,d^{-1}}$) when compared to TU-UP (DOC $=60\pm51$, Al $=1\pm1\,\mathrm{mg\,m^{-2}\,d^{-1}}$). In contrast, below the TU-UP profile, the mean biweekly fluxes of Ca and Mg ($205\pm143$, $59\pm39\,\mathrm{mg\,m^{-2}\,d^{-1}}$) are higher than under CU-UR (Ca $=77\pm62$, Mg $=13\pm10\,\mathrm{mg\,m^{-2}\,d^{-1}}$), except K flux, which is higher under the CU-UR profile ($137\pm98\,\mathrm{mg\,m^{-2}\,d^{-1}}$) compared to TU-UP ($31\pm24\,\mathrm{mg\,m^{-2}\,d^{-1}}$). The mean biweekly $HCO_3^-$ is higher under the CU-UR profile ($472\pm220\,\mathrm{mg\,m^{-2}\,d^{-1}}$) than under TU-UP ($300\pm278\,\mathrm{mg\,m^{-2}\,d^{-1}}$). At the 2A horizon, similar but no significant differences by vegetation oc-

cur for Ca, Mg, and K fluxes, except for DOC. In addition, there is no significant difference in Na and DSi fluxes between the vegetation types.

## 5 Discussion

### 5.1 Soil water dynamics under cushion-forming plants and tussock grasses

Overall, the good agreement between water fluxes from simulations and independent fluxmeter observations and between the observed and simulated soil moisture at three horizons indicates that the soil water processes are reasonably well represented by the HYDRUS-1D model. The outcomes of the model show that the soil water partitioning is different by vegetation type. The largest differences are attributed to the ETa, which is 1.7 times higher under cushion plants compared to tussock grasses (Table 3). During dry periods ETa under cushion plants is 2.5-fold higher than under tussocks. The higher solar radiation (1.3-fold) and wind speed (1.6-fold) during dry periods compared to wet periods (Table S7, Fig. S6) can promote conditions for higher evapotranspiration, as reported for alpine ecosystems (Knowles et al., 2015; Ochoa-Sánchez et al., 2020).

Soil moisture is decisive for the actual evapotranspiration (Vereecken et al., 2015). When evapotranspiration is not limited by water availability, the ETa can be ultimately controlled by the available energy (Quiring et al., 2015; Seneviratne et al., 2010). This is reflected in the relationship between ETa and ETp by vegetation type. Under cushion plants, the mean annual ETa (522 mm) is similar to ETp

**Table 5.** Mean (and 95 % confidence interval) of the annual solute fluxes ($\text{g m}^{-2}\,\text{yr}^{-1}$) for both study sites.

| Soil profile | CU-UR | | | TU-UP | | |
|---|---|---|---|---|---|---|
| Horizon | A | 2A | 2BC | A | 2A | 3BC |
| Ca | 1.82 | 1.26 | 0.32 | 6.20 | 3.04 | 2.40 |
| | [0.96–2.68] | [0.82–1.70] | [0.22–0.42] | [1.84–10.6] | [1.60–4.48] | [1.90–2.90] |
| Mg | 0.30 | 0.38 | 0.24 | 1.97 | 1.08 | 0.69 |
| | [0.18–0.42] | [0.22–0.54] | [0.16–0.32] | [0.29–3.65] | [0.04 –2.12] | [0.19–1.19] |
| Na | 1.59 | 1.62 | 0.68 | 2.52 | 2.05 | 2.63 |
| | [1.29–1.89] | [1.20–2.04] | [0.52–0.84] | [1.34–3.70] | [1.33–2.77] | [2.23–3.03] |
| *K* | 3.03 | 0.17 | 0.06 | 1.36 | 0.08 | 0.07 |
| | [1.95–4.11] | [0.09–0.25] | [0.02–0.10] | [0.00–3.80] | [0.06–0.10] | [0.05–0.09] |
| $HCO_3^-$ | 11.1 | 6.47 | 2.70 | 9.34 | 10.3 | 20.1 |
| | [5.66–16.5] | [3.57–9.37] | [2.16–3.24] | [4.44–14.2] | [7.52–13.0] | [16.1–24.1] |
| DOC | 17.4 | 2.53 | 0.36 | 1.47 | 1.58 | 1.95 |
| | [14.4–20.5] | [1.77–3.29] | [0.30–0.42] | [1.23–1.71] | [1.14–2.02] | [1.51–2.39] |
| Al | 0.62 | 0.04 | 0.01 | 0.03 | 0.03 | 0.02 |
| | [0.40–0.84] | [0.04–0.04] | [0.01–0.01] | [0.03- 0.03] | [0.01–0.05] | [0.00–0.04] |
| DSi | 5.95 | 5.76 | 3.29 | 7.25 | 8.70 | 9.61 |
| | [5.65–6.25] | [4.14–7.38] | [2.83 -3.75] | [4.49–10.0] | [7.16–10.2] | [9.21–10.0] |
| T.Cat. | 11.1 | 6.47 | 2.70 | 19.3 | 15.0 | 15.4 |
| | [5.66–16.5] | [3.57–9.37] | [2.16–3.24] | [13.2–25.5] | [12.5–17.2] | [13.1–17.7] |

T.Cat.: sum of Ca, Mg, Na, K, and DSi fluxes. CI: confidence interval = mean ± 95 % CI.

(526 mm), regardless of rainfall conditions (Tables 3, S7), indicating that there is water available for evapotranspiration during the whole study period. The strong decrease in soil moisture in the A and 2A horizons during the dry season and the steep increase in soil moisture during the first rainfall events after long dry periods (e.g., September 2019 and November 2020; Fig. 4a) are evidence of the high evapotranspiration and surface infiltration capacity in soils under cushion-forming plants. Under tussock grass, the soil moisture variation is less pronounced during dry periods and rainfall events (Fig. 4b): the mean annual ETa (308 mm) is well below the ETp (559 mm), and the transmission of water towards deeper soil horizons is high. The fact that the highest soil moisture is observed in the 2A horizon under tussocks is an indication of the higher soil water storage capacity of that layer. Under tussock grass at 95 cm depth, the mean annual soil moisture ($0.55\,\text{cm}^3\,\text{cm}^{-3}$) is higher than the soil water retention at field capacity ($0.54\,\text{cm}^3\,\text{cm}^{-3}$), indicating that the soil under tussocks is under saturation conditions at this depth over most of the study period.

The higher ETa under cushion-forming plants compared to tussock grasses is consistent with a recent study at the same sites based on water-stable isotopes (Lahuatte et al., 2022). The authors found that soils under cushion plants

are more prone to evapotranspiration than under tussock grass. This difference was attributed to the direct exposure of cushion plants' topsoil to solar radiation, whereas tussock grasses limit this exposure by producing a shadow effect with their leaves. The mean annual percentage of ETa in relation to rainfall below the tussock grass (43 ± 7 %) is slightly lower than in the southern Ecuadorian páramo grasslands (51 %; Carrillo-Rojas et al., 2019) and lower than other high-altitudinal alpine tundra in the USA (59 %; Knowles et al., 2015) and alpine meadows on the Qinghai–Tibetan Plateau in China (60 %; Gu et al., 2008). The mean annual percentage of ETa in relation to rainfall below the cushion-forming plants (76.0 ± 6.0 %) is higher than under páramo grasslands. This occurs despite the fact that the mean annual potential evapotranspiration (ETp) is 1.06 higher under tussock grass than under cushion plants (Table 3).

There is approximately 2-fold less water flux transmitted from the A horizon to the underlying horizons under cushion plants (Table 4, Fig. 7b). Likewise, deep drainage is also about 3-fold lower under cushion plants than under tussock grass (Table 3), especially during wet periods (Table S7, Fig. 6d). Deep drainage represents 19 % and 56 % of rainfall below cushion-forming plants and tussock grass, respectively. We attribute the differences in vertical water fluxes

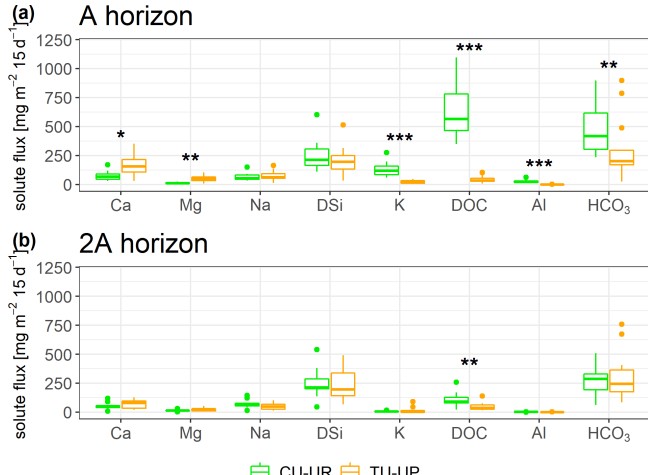

**Figure 9.** Biweekly solute fluxes in the two upper horizons of the CU-UR and TU-UP profiles. The Mann–Whitney $U$ test was applied for differences between vegetation types, with levels of significance *, **, and *** corresponding to $p<0.05$, $p<0.01$, and $p<0.001$.

and deep drainage to the difference in the vertical distribution of soil hydraulic functions (Figs. S8, S9). In the A horizon near or above field capacity (pF $\sim$ 2), high hydraulic conductivity and water retention under cushion-forming plants result in faster rainfall infiltration, higher water storage, and higher evapotranspiration compared to under tussock grasses (Figs. S8a, S9a). This results in the dynamic range in soil moisture in the A horizon under cushion plants, which reflects the filling and emptying caused by low-intensity rainfall and evapotranspiration (Fig. 4a). Under cushion plants, the coarser root system and lower BD in the A horizon result in higher $K_{\mathrm{SAT}}$ and water retention ($\theta_S$, $\theta_{\mathrm{FC}}$) compared to tussock grass (Fig. 3c, e), as previously reported (Páez-Bimos et al., 2022). In the 2A horizon under tussock grass, the water retention near field capacity (pF $\sim$ 2) is higher than in the A horizon and higher than under cushion plants in the 2A horizon (Fig. S8b). This results in higher soil moisture in the 2A horizon under tussock grass compared to the A horizon and the cushion plants in the 2A horizon (Fig. 4b). The hydraulic conductivity in the 2A horizon under both vegetation types lies on the same order of magnitude. In the 2BC horizon near field capacity, soil water retention for both vegetation types is in CE5 the same range; however, the hydraulic conductivity is higher under tussock grass compared to under cushion-forming plants (Fig. S8c, S9c). The latter allows for a continued infiltration of water below this horizon under tussock grass.

The simulation of the soil water balance shows that surface runoff would not be occurring, similar to what is observed in other regions characterized by high infiltration and low rainfall rates (Blume et al., 2009; Mosquera et al., 2022; Tobón and Bruijnzeel, 2021). Interception is considered negligible

for cushion plants since their leaves are placed directly on the ground (Fig. 2b–c). For tussock grass, we analyzed the interception loss influence by adapting a method developed for 100 % of vegetation cover (Ochoa-Sánchez et al., 2018). We reduced the vegetation cover to 55.8% (*Calamagrostis intermedia*; Table 1) as observed in the TU-UP profile to estimate the effective rainfall (rainfall − interception loss) and simulated the soil hydrological processes. This resulted in a 30 % reduction in effective rainfall that led to a severe reduction in simulated water content at the 2A and 3BC horizons, especially during the dry periods (Fig. S7). We consider that, by applying the adapted model for interception loss for the TU-UP profile, the rainfall reduction is overestimated since it eliminates most small rainfall events during dry periods. This is in line with the decreasing nonlinear interception rate in alpine grasslands when they lose vegetation cover (Genxu et al., 2012). Therefore, we consider that tussock canopy interception plays a minor role in the soil water balance of the TU-UP profile. The complete analysis of interception can be found in Sect. S4.

## 5.2 Solute fluxes under soil-forming cushion plants and tussock grasses

Bicarbonate and DSi were observed as the dominant soil solute fluxes, while the DOC flux was highest in the A horizon of the soil profile covered by cushion plants (Table 5, Fig. S5). The dominance of bicarbonate (2–21 g m$^{-2}$ yr$^{-1}$) and DSi (3–10 g m$^{-2}$ yr$^{-1}$) in the soil weathering fluxes is in line with the existing data on river chemical loads from páramo environments (Arízaga-Idrovo et al., 2022; Tenorio et al., 2018), where fluxes of 9 to 30 and 2 to 4 g m$^{-2}$ yr$^{-1}$ were reported for bicarbonate and DSi, respectively. For Andean basins, the solute fluxes show similarly high contributions of HCO$_3^-$, DSi, and also Ca (61, 13, and 18 g m$^{-2}$ yr$^{-1}$, respectively) (Moquet et al., 2016).

Vegetation type had a significant influence on solute fluxes in the upper A horizon, while the influence was lower and less significant in the lower 2A horizon (Fig. 9, Table 5). Annual DOC fluxes (0.4–3 g m$^{-2}$ yr$^{-1}$) are in line with the values reported from similar Andosols (0.9–8.4 g m$^{-2}$ yr$^{-1}$) as detailed in Table S11, except in the A horizon under cushion plants (17–23 g m$^{-2}$ yr$^{-1}$). The high DOC fluxes in the A horizon below cushion plants compared to tussocks can be attributed to a high DOC concentration (47 mg L$^{-1}$) rather than to water flux, since water flux is lower under cushions in the A horizon (Table 4). Besides cushion plants in the A horizon, mean DOC concentrations in our study (3.1–4.0 mg L$^{-1}$) are close to the range of values measured in Andosols grasslands: 4.7 mg L$^{-1}$ (by wick samplers, Pesántez et al., 2018) and 5–7 mg L$^{-1}$ (by lysimeters, Ugolini et al., 1988). DOC concentrations present a larger range when measured in soil solutions collected by lysimeters for Andosols under other vegetation covers (2–28 mg L$^{-1}$, Aran et al., 2001; Chen et al., 2017; Fujii et al., 2011). Extraordinarily high DOC concen-

trations (up to $60\,\mathrm{mg\,L^{-1}}$) have been reported for marshes, bogs, and swamps, related to large plant net primary productivity and slow-moving streams (Thurman, 1985).

We argue that the high DOC concentration under cushions in the A horizon can be partially explained by low downward water fluxes below the A horizon (Table 4, Fig. 7b) and by the filling and emptying of the soil water storage as a result of low-intensity rainfall (Padrón et al., 2015) and high ETa (Fig. 6c). The high DOC concentration is related to the high SOC ($9\,\%$) content in the A horizon under cushion plants that decreases strongly with depth (Fig. 3d). DOC might be attributed to SOC from roots/leaf litter decay (Kalbitz et al., 2000). High SOC and low pH ($<5$) in the rooting zone (Fig. 3d–g) under cushion-forming plants can point to a release of organic acids that contributes to the observed rise in DOC concentration (Molina et al., 2019; Takahashi and Dahlgren, 2016; Ugolini et al., 1988). For tussock grass, no consistent trends with depth can be found between DOC concentrations and any soil property.

The values of Al fluxes follow the behavior of DOC and Fe fluxes in both soil profiles. Under cushion plants, they decrease from the A horizon to the lower horizons, while, under tussock grass, they are lower but constant with depth (Table 5). DOC, Al, and Fe concentrations in soil solutes are significantly higher under cushion plants compared to tussocks at the 2A horizon (Fig. 8g, Table S8). This is also reflected in the higher DOC flux under cushion plants at the 2A horizon (Fig. 9b) despite much lower water fluxes. Leaching of DOC, Al, and Fe from the A horizon to the lower horizons has been reported in volcanic ash soils (Ugolini et al., 1988); however, we did not observe an accumulation of OC in the interface between the A and 2A horizons under cushion plants. Sampling DOC, Al, and Fe solutes at shorter time intervals and soil material at higher depth resolution could help to better understand the transport and remobilization of organic carbon below cushion plants.

Solute concentrations and fluxes of Ca, Mg, and K are influenced by the vegetation type in the A horizon (Fig. 9a, Table 5). At our sites, annual Ca fluxes ($0.3–5\,\mathrm{g\,m^{-2}\,yr^{-1}}$), Mg fluxes ($0.3–2\,\mathrm{g\,m^{-2}\,yr^{-1}}$), and K fluxes ($0.1–3\,\mathrm{g\,m^{-2}\,yr^{-1}}$) lie in the range of observed soil solute fluxes in other ecosystems (Table S11). Under tussock grass, Ca and Mg fluxes decrease with depth, while the same occurs for K under cushion plants (Fig. S5). The 3- to 5-fold higher fluxes of these nutrients in the A horizon can be attributed to the corresponding 3- to 5-fold higher element concentrations in the soil water solutions (Fig. 8a–b, e) rather than to the water fluxes (Table 4). The high concentrations and fluxes of Ca, Mg (for tussocks), and K (for cushions) in the near-surface horizons might reflect their uptake and cycling by plants (Amundson et al., 2007; Kelly et al., 1998; Lucas, 2001; White and Buss, 2014; White et al., 2009).

Vegetation type has a minimal but not significant effect on Na and DSi fluxes (Fig. 9, Table 5). Annual DSi fluxes at our sites ($4–9\,\mathrm{g\,m^{-2}\,yr^{-1}}$) are within the range of DSi fluxes in other ecosystems ($0–27\,\mathrm{g\,m^{-2}\,yr^{-1}}$; Table S11). The solute fluxes of DSi and Na follow closely the variation of water fluxes with depth (Table 5, Table 4). The increase in DSi with depth under both vegetation types can be related to the longer water transit time at depth: Lahuatte et al. (2022) estimated water transit times for the A horizons at $\sim 6$ months and for the 2A horizons at $\sim 1$ year. Bicarbonate concentrations seem to follow the vertical distribution of SOC content, root properties, and rooting depth, especially under cushion plants (Figs. 3d–f, 8h). Higher root and microbial respiration can enhance carbonic acid formation, which then dissociates into $HCO_3^-$ (Amundson et al., 2007; Perdrial et al., 2015; Ugolini et al., 1988). The fact that we observed the highest concentration of $HCO_3^-$ at the greatest depth (3BC) under tussock grass cannot be explained by the root network or SOC and needs further investigation (Fig. 8h).

### 5.3 The effect of vegetation on contemporary soil weathering through the soil water balance

Vegetation types associated with each soil profile showed differences in soil water balance and soil chemical weathering. Figure 10 depicts the conceptualization of our findings. The annual soil chemical weathering rates (T.Cat. fluxes) in this study ($5–17\,\mathrm{g\,m^{-2}\,yr^{-1}}$) are within the range of the values reported for cold environments ($2–3\,\mathrm{g\,m^{-2}\,yr^{-1}}$; Clow and Drever, 1996; Guicharnaud and Paton, 2006) and tropical forests ($22\,\mathrm{g\,m^{-2}\,yr^{-1}}$; White et al., 1998). The annual depth-integrated (sum over all horizons) soil chemical weathering under tussock grass ($46\,\mathrm{g\,m^{-2}\,yr^{-1}}$) is 1.5 times higher than under cushion plants ($30\,\mathrm{g\,m^{-2}\,yr^{-1}}$) (Table 5). This reflects mainly the higher water fluxes through the soil profile since solute concentrations for most elements are similar (Ca, Na, and Mg) or higher (K and DSi) under cushion plants (Fig. 8). This illustrates the higher dependence of soil weathering rates on the transmission of water in the soil profile than on the solute concentrations. The soil weathering rate is highest in the A horizon regardless of the vegetation type, showing the combined effect of higher water fluxes and higher Ca, Mg, and K concentrations. The soil weathering rate at the 2BC horizon corresponds to 0.4 times the rate at the A horizon under cushion plants, whereas under tussock grass the weathering rate at the 3BC horizon is 0.84 times the rate at the A horizon, resembling the decline in water fluxes with depth by vegetation type (Fig. 10a). This can also be linked to the progressive neutralization of soil solution acidity in the upper horizons that leads to less weathering with depth. This direct relationship between soil weathering rates and water fluxes at our sites corroborates the earlier work by Maher (2010). She found weathering rates that were strongly controlled by hydrological fluxes when the water residence times are between 5 d and 10 years and the water flow rates are below $16\,\mathrm{m\,yr^{-1}}$. Our data fall within this range, with soil water residence times of approximately 0.5 years to 1 year and water flow rates of 0.1 to $0.4\,\mathrm{m\,yr^{-1}}$.

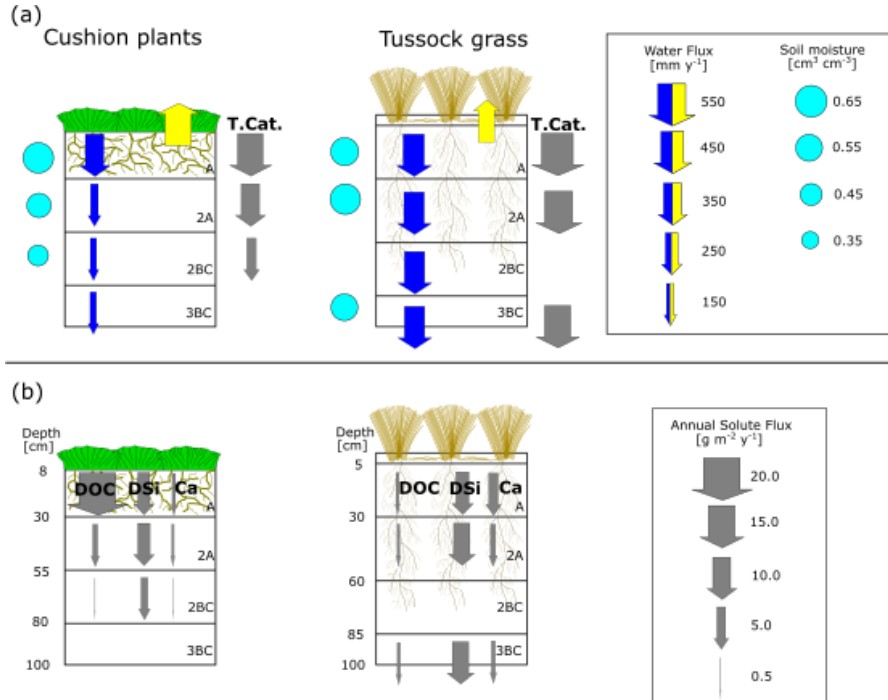

**Figure 10. (a)** Mean annual water fluxes, contemporary soil weathering rates, and soil moisture and **(b)** solute fluxes at different soil horizons under cushion-forming plants and tussock grass. The vertical arrows represent fluxes, of which size and direction indicate flux quantity and direction of flow. The blue and yellow arrows represent water fluxes: infiltration and ETa, respectively. The gray arrows represent soil chemical weathering rates (T.Cat.) and solute fluxes. Root depth, abundance, and diameter drawn as per Páez-Bimos et al. (2022).

Differences in soil hydraulic properties by vegetation type in the A horizon can influence soil water balance and soil chemical weathering for the entire soil profile (Fig. 3, Fig. 10). The higher ETa under cushion plants reduces substantially the mean annual water transmitted from A to the 2A and 2BC horizons (0.4 and 0.3 times, respectively), while under tussock grass, this decrease is almost negligible (Table 4). We argue that the coarser and shallow (up to 30 cm depth) root system under cushion plants is partially responsible for this difference in soil water balance by modifying soil structure and changing soil hydraulic properties. This is consistent with a recent theoretical study that relates the variability in evapotranspiration under a given climatic condition to the root architecture (Hunt, 2021). Several studies have also pointed to vegetation as an important influence on chemical weathering by altering water fluxes, storage, evapotranspiration, and soil water residence time (Brantley et al., 2017; Drever, 1994; Hunt, 2021; Kelly et al., 1998). Moreover, there is an intertwined and co-evolutionary relationship between infiltration and roots. Rooting depth has been associated with infiltration depth on a global scale and with soil water content at a local scale (Fan et al., 2017). On the other hand, rooting depth and root morphology can alter soil hydraulic properties and thus soil infiltration by changing the soil structure (Lu et al., 2020). At our study sites, shallower and coarser rooting under cushion plants resulted in lower

transmission of water below the A horizon and further lower soil chemical weathering rates, while under tussock grass a deeper and finer root system promoted steady water transmission with depth and higher soil chemical weathering.

In our study, soil water fluxes through the soil profile play a significant role in soil chemical weathering, while soil moisture measured in the field does not. In contrast, Cipolla et al. (2021) found a hysteric relationship between soil moisture and weathering rates in a short-term study and attributed this nonlinearity to a memory of past events (wet and dry periods). García-Gamero et al. (2022) found a positive relationship between soil moisture and soil weathering using a long-term model. We attribute this difference to the weak soil moisture seasonality in our study sites (Fig. 4), typical in tropical páramo ecosystems (Mosquera et al., 2022), which results in small ranges (max–min) of soil moisture for cushion plants ($\leq 0.09\,\mathrm{cm^3\,cm^{-3}}$) and tussock grasses ($\leq 0.07\,\mathrm{cm^3\,cm^{-3}}$) in comparison with larger soil moisture ranges from the modeled soil profiles ($0.35\,\mathrm{cm^3\,cm^{-3}}$) (Cipolla et al., 2021; García-Gamero et al., 2022).

## 6   Conclusions

We investigated the influence of soil–vegetation associations on soil water balance, solute fluxes, and contemporary soil

chemical weathering in two soil profiles under different vegetation types in the high tropical Andes. The soil water balance in the soil profile under cushion-forming plants (dominated by *Azorella pedunculata*) describes a two-layer system where the upper horizon stores water that is available for evapotranspiration and where water transmission to the horizons below is low. In contrast, the tussock grass profile (dominated by *Calamagrostis intermedia*) represents a homogeneous system that regulates and transmits water evenly through all the soil horizons. In our study sites, we found associations between root systems (related to vegetation type) and soil water balance and fluxes. Under cushion-forming plants, a shallower and coarser root system is related to a more porous soil structure that leads to higher total available water (storage) and higher saturated hydraulic conductivity (infiltration capacity) in the top horizon, while under tussock grass, a finer and deeper root system reflects less total available water and a lower but depth-constant saturated hydraulic conductivity, resulting in larger water fluxes being transmitted to lower horizons.

Vegetation type imposed a significant influence on solute fluxes in the upper A horizon, while the influence was lower and less significant in the lower horizons. Particularly high DOC, Al, and Fe fluxes are reported in the A horizon of soils under cushion-forming plants, despite relatively low water fluxes. Solute concentrations and fluxes of Ca, Mg, and K are 3- to 5-fold higher in the A horizon and differ by vegetation type, which can point to differences in plant biogeochemical fluxes between cushion-forming plants and tussock grasses. Other solutes like DSi and Na are only minimally influenced by vegetation type.

Chemical weathering rates are more imprinted by the soil water fluxes than by the solute concentrations. In the young volcanic ash soils from the high Ecuadorian Andes, contemporary soil chemical weathering rates differ by vegetation type, as the vegetation modifies the soil hydraulic properties in the upper horizon, which in turn results in changes in the soil water balance. This shows the importance of considering soil water fluxes when investigating the effect of hydrological conditions on soil weathering. Our findings reveal the role of the vegetation type in modifying the soil biogeochemical and hydrophysical properties of the uppermost soil horizon and, hence, in controlling water balance, solute fluxes, and contemporary soil chemical weathering throughout the entire soil profile.

*Data availability.* The data that support this study are available from the ParamoSus project upon reasonable request.

*Supplement.* The supplement related to this article is available online at: https://doi.org/10.5194/hess-27-1-2023-supplement.

*Author contributions.* Conceptualization: VV, AM, SPB; experimental design: VV, AM, PD; data collection: SPB, MC, BL, MV; data validation: SPB, BL, MC; funding acquisition: VV, MV, PD; project administration: VV, MV; writing – original draft: SPB, VV; writing – review and editing: all the authors.

*Competing interests.* The contact author has declared that none of the authors has any competing interests.

*Acknowledgements.* We thank Jordan Cruz, Isaías Quinatoa, and other undergrad students from the Escuela Politécnica Nacional (EPN) and Universidad Central del Ecuador for their help in the preparation and sampling of field campaigns in Antisana. We were supported by TIGRE Ecuador in providing the PVC tube for soil water content reflectometer calibration. We thank the Centro de Investigaciones y Estudios de Ingeniería de los Recursos Hídricos (CIERHI) y and the Centro de Investigaciones y Control Ambiental (CICAM) at the EPN for access to computers for hydrological simulations and for supporting field chemical analyses. This research has been supported by the research cooperation project "Linking Global Change with Soil and Water Conservation in the High Andes" (ParamoSus) funded by the Académie de Recherche et Enseignement Supérieur de la Fédération Wallonie-Bruxelles (ARES), Belgium. Sebastián Páez-Bimos received a granting CE6 license from the EPN for the development of his doctoral program.

*Financial support.* This research has been supported by the Académie de Recherche et Enseignement Supérieur (ARES–PRD–ParamoSus). TS4

*Review statement.* This paper was edited by Nadia Ursino and reviewed by two anonymous referees.

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

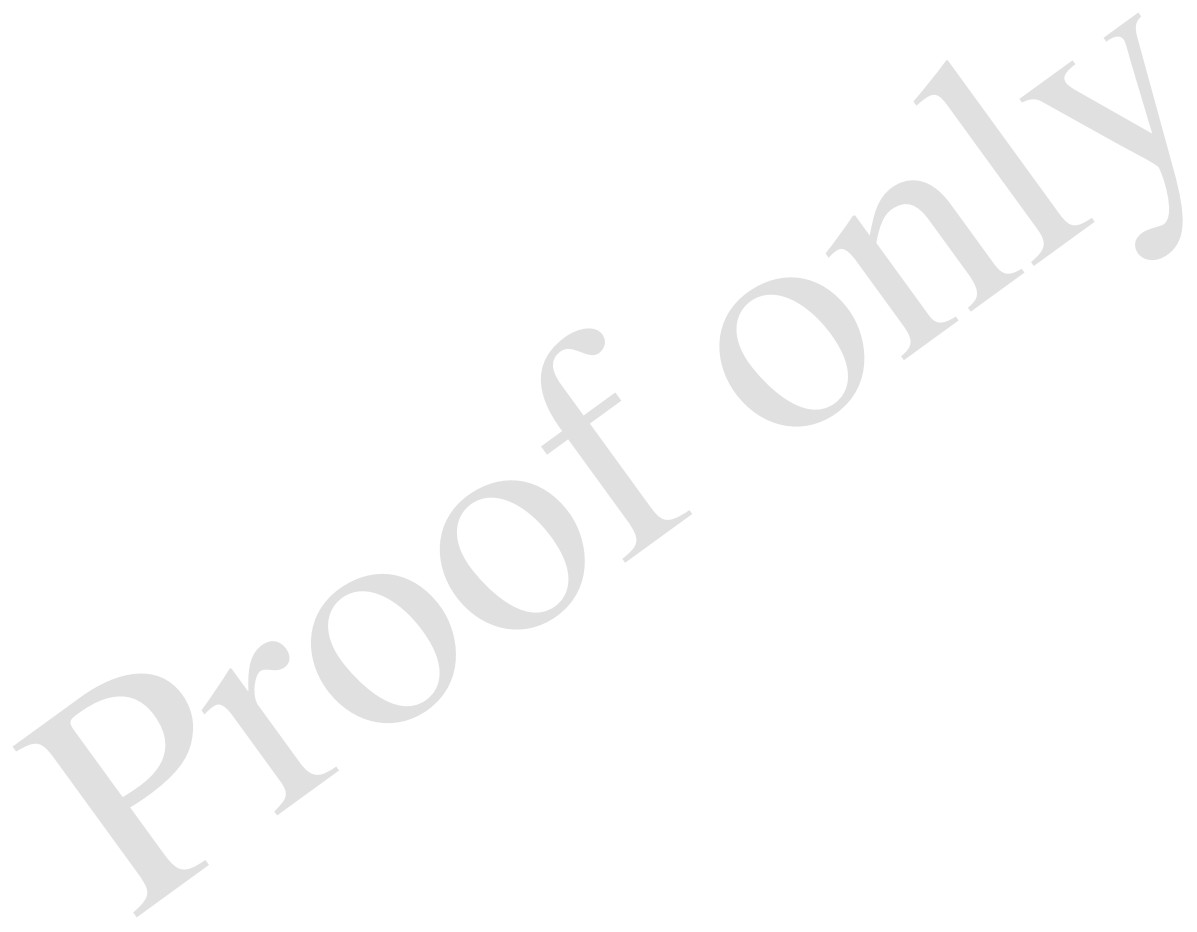

**Remarks from the language copy-editor**

CE1     Please provide a metric equivalent value here.
CE2     Please define this abbreviation here.
CE3     Please confirm.
CE4     "1.06 times"?
CE5     Please confirm.
CE6     Just "grant"?

**Remarks from the typesetter**

TS1     Please use consecutive numbers for the tables and figures in your Supplement. Supplementary material 2: Please change Figure 1 to Fig. S10. Supplementary material 4: Please change Figure 1 to Fig. S11, Figure 2 to Fig. S12, and Figure 3 to Fig. S13. Please also change Table 1 in Supplementary material 2 to Table S12. Finally, please rename the headings: Supplementary material 1 to Section S1, Supplementary material 2 to Section S2, Supplementary material 3 to Section S3, Supplementary material 4 to Section S4 and send us the new Supplement PDF. Many thanks.

TS2     The composition of Figs. 5 and 9 has been adjusted to our standards.

TS3     Please note that units have been changed to exponential format throughout the text. Please check all instances.

TS4     Please note that there is a discrepancy between funding information provided by you in the acknowledgements and the funding information you indicated during manuscript registration, which we used to create this section. Please double-check your acknowledgements to see whether repeated information can be removed from the acknowledgements or changed accordingly. If further funders should be added to this section, please provide the funder names and the grant numbers. Thanks.

TS5     Please provide more information like a persistent identifier and publisher.

TS6     Please ensure that any data sets and software codes used in this work are properly cited in the text and included in this reference list. Thereby, please keep our reference style in mind, including creators, titles, publisher/repository, persistent identifier, and publication year. Regarding the publisher/repository, please add "[data set]" or "[code]" to the entry (e.g. Zenodo [code]).

TS7     Please provide a persistent identifier (ISBN or DOI).

TS8     Please provide volume.

TS9     Please provide page range or article number.

TS10     Please provide a persistent identifier (ISBN or DOI).

TS11     Please provide volume and page range or article number.

TS12     Please provide a persistent identifier (ISBN or DOI).

TS13     Please provide the publisher and a persistent identifier (ISBN or DOI).

TS14     Please provide the publisher and a persistent identifier (ISBN or DOI).

TS15     Please provide page range or article number.

TS16     Please provide a persistent identifier (ISBN or DOI).