# Peer review of "Soil-vegetation-water interactions controlling solute flow and chemical weathering in volcanic ash soils of the high Andes"

_Hydrology and Earth System Sciences, 2022_

## Community Comment (CC1)

**Review report on "Soil-vegetation-water interactions controlling solute flow and transport in volcanic ash soils of the high Andes"**

by Sebastián Páez-Bimos, Armando Molina , Marlon Calispa, Pierre Delmelle, Braulio Lahuatte, Marcos Villacís, Teresa Muñoz and Veerle Vanacker

| | |
|---|---|
| Author: | Esther Geertsma |
| Journal: | Hydrological Earth System Sciences Discussions |
| Date: | 27/10/2022 |

The paper "Soil-vegetation-water interactions controlling solute flow and transport in volcanic ash soils of the high Andes" by Páez-Bimos et al. investigates the influence of two different types of vegetation (cushion plants and tussock grasses) on soil water balance, solute fluxes and chemical weathering in the high Andes ecosystem via fieldwork and numerical modelling. The cushion plants had lower water transmission below the A horizon and lower soil chemical weathering rates due to their shallower and coarser roots. The tussock grasses had steady water transmission throughout the soil profile and higher soil chemical weathering due to their deeper and finer roots. The paper concluded the soil water balance under the two vegetation types was different due to the root structure of the plants, which altered solute fluxes and soil chemical weathering throughout the depth of the soil.

Studies done in the past about the volcanic ash soils of the high Andes focused mostly on land use (Podwojewski et al., 2006; Buytaert et al., 2002; Buytaert et al. 2007) and on soil properties (Buytaert et al., 2005; Zehetner et al., 2003; Tonneijck et al., 2010; Tonneijck et al., 2010; Zehetner & Miller, 2006), while more recent studies in the area focused on its vegetation effects on hydraulic soil properties (Páez-Bimos et al., 2022), soil hydrology (Lahuatte et al., 2022), runoff processes (Minaya et al., 2021), and weathering (Barbosa et al., 2022). This paper is in line with these topics, since it focuses on vegetation, weathering, and hydrology. It is most comparable with Páez-Bimos et al. (2022) by also researching vegetation effects on hydraulic soil properties. Some of their conclusions were repeated in this paper, for example:
- Compared to tussock grasses, the higher water retention capacity at saturation under cushion-forming plants can enhance soil water storage during prolonged rainfall events, whilst the higher total available water results in higher water storage for plants and can promote evapotranspiration during the dry season.
- The saturated hydraulic conductivity in the top horizon is higher under cushion forming plants.
- Below the rooting zone, the saturated hydraulic conductivity drops remarkably, especially under cushion plants.

However, I think there are sufficient differences between the two papers. Páez-Bimos et al. (2022) touches upon the soil pore structure and the specific plant root characteristics that were found to be notably different between the two vegetation types, which is not included in this paper. This paper focuses on solute fluxes and chemical weathering, which is not discussed in the paper by Páez-Bimos et al. (2022).

According to the paper, vegetation effects on individual components of the soil water balance have been studied before. Furthermore, Molina et al. (2019) found that there are significant differences in soil chemical weathering between vegetation patterns. This paper aims to reveal the mechanics behind vegetation influencing soil weathering rates via its root system effects on soil water fluxes,

since this has not been evidenced before. It adds knowledge about soil-vegetation-water interactions that may be relevant for soil hydrology, soil chemistry and ecology fields. Therefore, I think the paper addresses relevant scientific questions within the scope (advancing understanding of hydrological systems) of the journal Hydrology and Earth System Sciences well.

The paper is well written and uses understandable and precise language. It includes figures that support the understanding of, summarize, and add to the text well. Especially figure 10 helps to understand and summarize the text very well. The root diameter and deepness are visualized here, which puts their differences into perspective. The differences in arrow sizes are distinguishable and immediately show the results in a glance. However, I do have some issues – mostly with the methods and conclusions – that I would like the authors to address before publishing. Besides the strengths of the paper, I will elaborate on these issues below.

I think your introduction is well written and gives a good background and understanding for the topic of the paper. It also includes relevant references. I think it creates a nice funnel from a broad perspective (the relationship between soil hydrology and chemical weathering and how this has been researched before; the relationship between vegetation, soil hydrology and weathering; and the Andes ecosystem) to the research questions and problem statement. My personal preference would be to move the information on the Páramo ecosystem (line 99-109) from the introduction to the site description in the methods (paragraph 3.1), but this is entirely up to your own preferences.

From the methods section in your paper, I understood almost everything that you did. Everything was written down very clearly and most already existing methods were referenced properly (e.g: the undisturbed samples by the multi-step apparatus (van Dam et al., 1994) in line 157-158; "... by the Eijkelkamp pressure membrane apparatus (Klute, 1986)" in line 160; and "... using the two heads method (5 and 10 cm) (Reynolds and Elrick, 1985)" in line 165-166. Aso, figures 1d and 1e helped visualize the vegetation types and table 1 helped understand the text by visualizing what these sample locations look like. However, the reasoning behind some of your methods are not fully clear to me. For example:
- It is unclear from the text that you did not violate the assumption of independence between observations of the Mann Whitney U test since you took multiple samples from one location. Can you prove that you did not violate this assumption?
- Also, you write that the mean annual rainfall values (2019-2020) did not differ significantly between the two sites (line 375). You however do not show what statistical test is used to determine this significant difference. Can you include this in your methods?
- Furthermore, you write that, for preparing the soil samples for soil texture determinations, the carbonates and OM were removed. Bieganowski et al. (2018) writes that, while there is no uniform soil preparation standard compiled, the best known method describes that – besides OM – soluble and gypsum removal is obligatory, and besides carbonate, iron oxide removal is optional. Can you argue why you only removed the carbonate and OM and not these other components?
- In your introduction, you reference Molina et al. (2019): "*Here, we take advantage of the mosaic-like distribution of vegetation types in the high Andes ecosystem, changing over short distances and allowing other factors (i.e., climate, geology, soil age, and topography) to remain constant*". This sentence makes it seem like it is important for the topography to remain constant for this experiment. However, Molina et al. (2019) write "*High Andean tropical ecosystems provide a good opportunity to study the association between chemical weathering, local topography, and vegetation patterns: the climate, parent material, and soil age can be held constant at the landscape scale, while the vegetation and slope morphology*"

*can vary greatly from the hilltops to the valley bottoms.*". Therefore the topography can vary between the two locations. Another paper done at your study sites (Páez-Bimos et al., 2022) concludes "*Soil hydraulic properties and soil pore structure changed in the uppermost horizons (A1 and A2) under cushion-forming plants and tussock grasses; whereas they did not change at topographic position.*". Can you specify whether the topography between the two sites is the same and if it was not, whether this influences your conclusions?

To conclude, while I think your scientific methods are clearly outlined and reproduceable. I think by elaborating more on these methods, your approach will have a more solid basis for the reader.

I like that you explain your results in the discussion with many references, e.g.: "The higher ETa under cushion-forming plants compared to tussock grasses is consistent with a recent study in the same sites based on water stable isotopes (Lahuatte et al., in rev.)" in line 483-484; and "The limitation of soil water transmission (e.g., by a reduction in vertical hydraulic conductivity) can result in saturation even at low rainfall intensities during prolonged precipitation events (Burt and Butcher, 1985). Thus, under tussock grass in the A horizon, the soil water storage capacity is limited by..." in line 497-499. However, your conclusion that a shallower and coarser root system is related to a more porous soil structure is not explained anywhere in the text. I think you should explain this in your discussion. I would recommend to reference to your previous paper Páez-Bimos (2022), possibly with some of the other results from there, like relating the strong decrease in saturated hydraulic conductivity with depth under cushion forming plants facilitating the soils to become saturated faster to the ash deposits.

In your paper, you make some conclusions that I could not find significant evidence for in your paper. I think this weakens the conclusions of the paper: right now I am not convinced that the conclusions are substantiated. For example:
- In the conclusion (line 669), you write "*Other solutes like DSi, Na, HCO₃⁻ are only minimally influenced by vegetation type.*". However, the highest $HCO_3^-$ concentration was found at the greatest depth under tussock grass which, according to the you, cannot be explained by the root network or SOC and needs further investigation (line 596). Therefore it is unclear to me how you reach the conclusion that there is only a minimal influence from vegetation type on $HCO_3^-$ if the highest found concentration cannot be explained by the root network or SOC and needs further investigation. You now only include sources about $HCO_3^-$ (in line 595) that found "higher root and microbial respiration can enhance carbonic acid formation, which then dissociates into $HCO_3^-$". Can you reference a source that finds the deep high $HCO_3^-$ concentration cannot be explained by the root network or SOC?
- You conclude "*In our study sites, we evidenced the role of root systems in regulating the soil water balance.*" (line 668-669). I did find statistical evidence that there were differences between the vegetation types in solute concentrations and fluxes, but I could not find statistical evidence that there were differences between vegetation types in the soil water balance or evidence for this causality. Can you explain how you proved the causality of vegetation root systems regulating the soil water balance?
- The annual solute fluxes of cations and DSi (taken as a proxy for weathering) are systematically higher in the TU-UP profile (line 449-451), but you write Na and DSi differences are not significant (line 589, although figure 8 shows Na differs significantly at 20 and 40 cm and DSi differs significantly at 40 cm) even though DSi was observed as one of the two dominant soil solute fluxes (line 538). Can you explain how you evidenced that there are chemical weathering differences between the vegetation types if this is the case?

- I cannot find statistical evidence for the causality relationship of the vegetation type affecting the contemporary soil chemical weathering (line 673-674). Can you prove that there is causality here?

In conclusion, I do not think the statistics and results you showed are sufficient to support your conclusions at this moment. I think including some sources for $HCO_3^-$ and evidence of causality will prove you reached substantial conclusions, which would improve the paper remarkably.

I also have some minor issues, that are summed up below:
- I am not sure if the title clearly reflects the contents of the paper. First of all, it does not become clear in the paper what the difference between solute flow and transport is. Also, the title is only associated with research question ii. There is no mention of chemical weathering, which was addressed in research question iii.
- The abstract only mentions that the vertical distribution of soil properties associated with the root systems. It does not explain what the mechanism is here, while the role of the roots is explained in the conclusion quite well. I think this misses from the abstract. Also, the abstract does not mention the location of the measurements.
- Your paper aims to fill the knowledge gap of how vegetation can influence contemporary weathering rates through its effect on soil water fluxes and transport. It is unclear how this information can be applied, can you give an argument for why the knowledge gap needs to be filled?
- In figure 4, the dots are very hard to distinguish from the simulated line. Also, the legend colors are very hard to relate to the colors in the picture.
- I think the number and quality of your references is appropriate. When I checked some of your sources (e.g. Fan et al., 2017; Jiang et al., 2018), I could not find any incorrect interpretations and these sources seemed of high scientific quality. However, I noticed that you included two papers in your reference list that were under revision by the time you wrote the paper. When I checked them it appeared that they have been published already. I will include them (Lahuatte et al., 2022; Páez-Bimos et al., 2022) in my reference list for you to include them.
- Páez-Bimos (2022) quantifies the root abundance and diameter, is that what you used for the relative root sizes in figure 10? If so, can you reference this paper in your figure for that?
- The paper is inconsequent in the APA references in the text: when two authors are included, sometimes '&' is used, sometimes 'and' is used.
- P7, line 187 and p9, line 235 reference to figure 2e, which does not exist.
- P7, line 160 writes ""36 undisturbed and 18 undisturbed samples". I think one of these should be "disturbed".
- P26, line 619 says "0.4 times, respectively". However, I am not sure this is correct since the two transmissions do not both seem 0.4. When I did my own calculation it seemed more like "0.44 and 0.79 times respectively"?

**References:**

Barbosa, A. M., Francelino, M. R., Thomazini, A., Schaefer, C. E. G. R., Anjos, L. H. C., Pereira, M. G., & Lyra, G. B. (2022). The thermal regime and mineralogical attributes of highland volcanic-ash soils from the Cotopaxi volcano, Ecuador: Absent permafrost and little pedogenesis. Geoderma Regional, 29, e00496.

Bieganowski, A., Ryżak, M., Sochan, A., Barna, G., Hernádi, H., Beczek, M., ... & Makó, A. (2018). Laser diffractometry in the measurements of soil and sediment particle size distribution. Advances in agronomy, 151, 215-279.

Buytaert, W., Deckers, J., & Wyseure, G. (2007). Regional variability of volcanic ash soils in south Ecuador: The relation with parent material, climate and land use. Catena, 70(2), 143-154.

Buytaert, W., Deckers, J., Dercon, G., De Bievre, B., Poesen, J., & Govers, G. (2002). Impact of land use changes on the hydrological properties of volcanic ash soils in South Ecuador. Soil use and management, 18(2), 94-100.

Buytaert, W., Sevink, J., De Leeuw, B., & Deckers, J. (2005). Clay mineralogy of the soils in the south Ecuadorian páramo region. Geoderma, 127(1-2), 114-129.

Fan, Y., Miguez-Macho, G., Jobbágy, E. G., Jackson, R. B., and Otero-Casal, C.: Hydrologic regulation of plant rooting depth, Proc. Natl. Acad. Sci. U.S.A., 114, 10572–10577, https://doi.org/10.1073/pnas.1712381114, 2017.

Jiang, X. J., Liu, W., Chen, C., Liu, J., Yuan, Z.-Q., Jin, B., and Yu, X.: Effects of three morphometric features of roots on soil water flow behavior in three sites in China, Geoderma, 320, 161–171, https://doi.org/10.1016/j.geoderma.2018.01.035, 2018.

Lahuatte, B., Mosquera, G. M., Páez-Bimos, S., Calispa, M., Vanacker, V., Zapata-Ríos, X., ... & Crespo, P. (2022) Delineation of water flow paths in a tropical Andean headwater catchment with deep soils and permeable bedrock. Hydrological Processes, e14725.

Minaya, V., Camacho Suarez, V., Wenninger, J., & Mynett, A. (2021). Runoff generation from a combined glacier and páramo catchment within the Antisana Reserve in Ecuador. Journal of Ecohydraulics, 1-16.

Molina, A., Vanacker, V., Corre, M. D., and Veldkamp, E.: Patterns in soil chemical weathering related to topographic gradients and vegetation structure in a high Andean tropical ecosystem, Journal of Geophysical Research: Earth Surface, 124, 666–685, 2019.

Páez-Bimos, S., Villacís, M., Morales, O., Calispa, M., Molina, A., Salgado, S., ... & Vanacker, V. (2022). Vegetation effects on soil pore structure and hydraulic properties in volcanic ash soils of the high Andes. Hydrological Processes, 36(9), e14678.

Podwojewski, P., Poulenard, J., Zambrana, T., & Hofstede, R. (2002). Overgrazing effects on vegetation cover and properties of volcanic ash soil in the páramo of Llangahua and La Esperanza (Tungurahua, Ecuador). Soil Use and Management, 18(1), 45-55.

Tonneijck, F. H. (2006). Volcanic ash soils in Andean ecosystems: unravelling organic matter distribution and stabilisation. Radiocarbon, 48(3), 337-353.

Tonneijck, F. H., Jansen, B., Nierop, K. G. J., Verstraten, J. M., Sevink, J., & De Lange, L. (2010). Towards understanding of carbon stocks and stabilization in volcanic ash soils in natural Andean ecosystems of northern Ecuador. European Journal of Soil Science, 61(3), 392-405.

Zehetner, F., & Miller, W. P. (2006). Erodibility and runoff-infiltration characteristics of volcanic ash soils along an altitudinal climosequence in the Ecuadorian Andes. Catena, 65(3), 201-213.

Zehetner, F., Miller, W. P., & West, L. T. (2003). Pedogenesis of volcanic ash soils in Andean Ecuador. Soil Science Society of America Journal, 67(6), 1797-1809.

---

## Author Comment (AC1)

**Author comments (ACs)**

The authors thank the reviewers for their constructive comments. The comments are shown in black font, and our responses are in regular blue font.

**RC1: 'Comment on hess-2022-294', Anonymous Referee #1, 22 Oct 2022**

In general, I found the paper to be well written and the study well-designed and carried out.

I found – however – some points to be limited that needs to be addressed. First, the research gap was not clarified and the presented hypothesis was already addressed in other studies (see comments below). Some additional explanation on why the approach observation/modeling was used, would help the readers to capture the study design and idea early on in the work. I also found that the modeling was no explained with the necessary detail and should be accompanied by an uncertainty assessment.

We respond in detail below.

Line 50 "the water balance"

The word "the" will be included

L60: what about porosity and soil particle surface area?

We will analyze the effect of porosity and soil particle surface area on weathering depth gradient.

The break from L62 to L64 is quite harsh.

We will include a transition sentence in the line 64 vegetation role on soil weathering.

L89 and 90: We already know that the hypothesis is true. It has been shown in numerous studies before. The paper would greatly benefit to make the research gap more clear and have the hypothesis and questions clearly linked to that. While the intro gives a good overview of what has been done, the research gap is only vaguely noted "Soil vegetation-water interactions are not fully understood". After that, the authors summarize things that are known, and not what is wrong in our current status quo. Points that are made e.g. in L116, should be in the introduction. This applies in general for section 2. Points that are made related to the research gap need to move into the intro.

We will sharpen this part and make the research gap clearer. While section 1 gives an overview of SoA and knowledge gaps in the field of soil ecohydrology and biogeochemical weathering, section 2 is more specific for the paramo ecosystem. We prefer to keep this structure, with section 2 focusing more on specificities of the paramo ecosystems

An explanation on the methodological choices would be helpful, which type of measurement supports which question and for what is the modelling needed?

We will add a sentence in the intro that relates the measurement and modelling to the research questions.

L123-125: not needed/relevant

We will delete the first sentence "The páramo ecosystem supplies drinking water to more than half a million inhabitants of Ecuador's capital, Quito (EPMAPS and FONAG, 2018)."

L215: what are the sensitive parameters? Do you enforce that the relation between ksat from one depth to the other is retained? What parameter ranges were used for the inverse simulation? Why? In general, 3.4. lacks the necessary details to reproduce or understand the simulation setup.

The details of the modelling are included in the Supplementary material 3. We will include details of sensitive parameters and the inverse simulation in the text.

Fig. 3. I guess the horizontal grey bars indicate the boundaries of a horizon. Yet, this needs to be explained in the captions and not left for guessing. CU-UR and TU-UP should also be explained in the caption or written out to make the figure stand-alone from the text.

We will include the explanation of the horizontal gray bars in the caption of Figure 3.

L309ff. Do the authors have any idea on the general variation of these soil properties beyond the two profiles? I am wondering if the difference is random or if this is really an effect/linked to the vegetation. I understand that the sampling is laborious, but I guess we all know sites where ksat does change by several orders of magnitude on very small spatial differences. Even though I agree that the A horizons of the sites seem to be quite different.

We observed the same distribution of soil properties in six other soil profiles described in Páez-Bimos et al., 2022, and in the larger study area that was surveyed for soil mapping by Erauw (2019). We will include a sentence indicating that the variation of soil properties is associated to vegetation types.

Páez-Bimos, S., Villacís, M., Morales, O., Calispa, M., Molina, A., Salgado, S., ... & Vanacker, V. (2022). Vegetation effects on soil pore structure and hydraulic properties in volcanic ash soils of the high Andes. Hydrological Processes, 36(9), e14678.

Erauw A. 2019. Soil horizon thickness as indicator of soil production and transport along slopes. Master thesis, Université catholique de Louvain, Louvain-la-Neuve, Belgium.

L327: What are the calibrated values and how do they differ between the profiles and to the measured values? What are the "sensitive paramters" (L.215).

We will include in this section the sensitive parameters and the calibrated values. We will compare them in relation to the measured values.

L333: I would partly disagree. When you have a KGE of 0.08, you barely explain anything of the observed behavior. So what is the problem? What could be the problem? Preferential flow? Also, a full uncertainty analysis of the simulation should be added rather than 3 simulations in Figure 4. Furthermore, I am having a hard time to distinguish the different lines on the plot.

We have reworked the manuscript, and have done a full uncertainty analysis. We will also compare the KGE value with reference values given in Knoben et al. (2019).

Knoben, W. J., Freer, J. E., & Woods, R. A. (2019). Inherent benchmark or not? Comparing Nash–Sutcliffe and Kling–Gupta efficiency scores. Hydrology and Earth System Sciences, 23(10), 4323-4331.

L521: Replace "The soil hydrology' simulations" by "The simulation of the soil water balance"

We will replace it

L558: Can you estimate the residence time? Or the general difference between your sites?

We will remove the part related to residence time

L585ff. In this section, mostly literature is cited, however it would be more straightforward to argue from your observation rather than relying on a reference. Of course, other work can then be references.

We take this point, and will rewrite this section and argue directly from our observations. This will be done in section 5.2 from L585 onwards.

---

## Author Comment (AC2)

**Author comments (ACs)**

The authors thank the reviewers for their constructive comments. The comments are shown in black font, and our responses are in regular blue font.

**RC2: 'Comment on hess-2022-294', Anonymous Referee #2, 15 Nov 2022**

This manuscript evaluates soil water and solute fluxes in two soil profiles under two different vegetation, i.e., the cushion-forming plants (CU-UR) and tussock grasses (TU-UP). This evaluation was based on measurements of soil water content, water flux density, and solute concentrations and the outputs from the well-known HYDRUS-1D model. These measurements and modeling allowed for evaluating the role of soil water balance on soil chemical weathering, which was one of the paper's main objectives.

Overall, the paper is well-written and suits the scope of HESS. Nevertheless, essential points must be included or further analyzed, especially in the methodology and discussion sections. These major points are described below, and a few minor considerations are described afterward.

We respond in detail below.

In the introduction, the statement "we hypothesize that vegetation type has an impact on water and solute fluxes (...)'' does not suit well as a  hypothesis since it has been well demonstrated, especially in the context of root water uptake modeling. Therefore, the hypothesis should be stated just as a particular case for these two vegetation types, not as an overall case.

We will adjust the hypothesis for our particular case

There needs to be more information about how ETa and ETp were determined. Only lines 229-230 say that "ETa was derived from potential evapotranspiration (ETp) according to the surface pressure head and soil moisture'' and in the supplementary material 3 it is stated that "... ETp [was] based on the Penman-Monteith equation''. Notice that ETp from the Penman-Monteith method requires values of parameters such as crop   canopy resistance and albedo. What were the values for each vegetation type? Furthermore, ETp is usually partitioned into potential transpiration and soil evaporation in hydrological modeling. Therefore, the authors should also shortly describe this and present the values of the parameters for each vegetation.

We will include more details on the determination of the ETp and how we used the Penman-Monteith equation in this area. There are very few data on the plant transpiration and soil evaporation in these environments, and even less on the potential impact of vegetation roughness and albedo on evapotranspiration. The focus of this study is the water flux modelling, not that much the evapotranspiration modeling.

The methodology lacks information about root measurements, yet Figure 3 shows the vertical distribution of root diameter and abundance. Only the maximum rooting depth for each vegetation is given (lines 133--134). The authors also need to show how they determined the relative distribution of root length density over depth. How was it considered in the HYDRUS model? What about the transpiration reduction function? As far as I know, Hydrus-1D has two options for transpiration reduction functions: the Feddes and Jarvis model. I would like to know about values used for the empirical threshold parameters and if they can impact the simulation results.

We considered that the effect of the root system is included in the measured soil properties. As indicated above, we did not consider the partitioning of evapotranspiration into evaporation and transpiration. It was not the goal of this manuscript. We will provide more details on the root measurements in the revised manuscript.

At the beginning of section 5.1, the authors compare ETa from CU-UR and TU-UP and cite values from other studies(in the second paragraph) but do not explain why ETa from CU-UR is higher than ETa from TU-UP. This difference is enhanced in dry periods but is also not explained. This discussion is brought back only in the fourth paragraph. First, in my opinion, this discussion should be placed right after the first paragraph. Second, the authors show some evidence for the higher (lower) ETa from CU-UR (TU-UP) but need to address why this happens. For instance, the authors should explain why in 2A horizon under tussocks, which has about 50% of the roots (roughly looking at Figure 3f ), the highest soil moisture is observed, but annual ETa is well below ETp. Overall, the discussions are mainly based on soil water content, but when it comes to actual transpiration, one should look at the soil pressure head. Thus, the differences in soil hydraulic functions between soil layers and the two soil profiles need to be considered in the discussions. Papers related to root water uptake modeling might be useful to enhance these discussions.

We can reorganize the text to improve the flow of the text. We will place the fourth paragraph at the beginning of section 5.1. We will further look into the differences in the soil hydraulic functions between soil horizons and profiles, and analyze how they are related to differences in ETa and ETp. We will include in the discussion of section 5.1 the differences in fitted water retention curves.

The discussion about soil water flux needs further analysis (section 5.1, 494--506). The authors attribute the differences in vertical water fluxes and deep drainage to the vertical distribution of soil water storage. However, notice that soil water content distribution is also affected by water flux. Also, stating that "soil water storage capacity is limited by lower $\theta_T$ AW and KSAT'' is misleading since a high hydraulic conductivity promotes the reduction of soil water content in the soil layer. I think this discussion should be based on the soil hydraulic functions from each soil profile, considering the hydraulic conductivity and soil retention curve rather than on soil water content or soil water storage.

Many thanks for this suggestion, we will include in the revised discussion of section 5.1 the differences in fitted water retention curves.

Minor comments:

1) l187. There is no Figure 2e.

We will change 2e with 2d.

2) l218. The Kling-Gupta efficiency (KGE) may not be well known as the other measures for goodness-of-fit and may need some description and citation.

We will include additional information on KGE

3) l228. What do you mean by surface pressure head?

We will explain this sentence in more detail

4) l334--336. These relations are not clear to me. Can we see them in any table?

The parameters are in the Supplementary material but we can include these tables in the main text

5) l410. Measured or simulated values in Table 3?

We will clarify this in the Table 3 description

7) Fig 7. The line colors for each soil horizon are hard to recognize in the figure.

We will modify the colors of the lower horizons.

6) l494. ``There is approximately 3-fold less water flux transmitted from the A to the underlying horizons under cushion plants (Table 3, Fig.    7b)''. It does not match to what is shown in Table 3.

We will rephrase the sentence

7) The Equation 1 from the supplementary material 3 must have the sink term for root water uptake.

We did not include the sink term for root water uptake since we are not considering the partitioning of the evapotranspiration into evaporation and transpiration.

---

## Author Comment (AC3)

**Author comments (ACs)**

The authors thank the reviewers for their constructive comments. The comments are shown in black font, and our responses are in regular blue font.

**CC1: 'Comment on hess-2022-294', Esther Geertsma, 07 Nov 2022**

The paper "Soil-vegetation-water interactions controlling solute flow and transport in volcanic ash soils of the high Andes" by Páez-Bimos et al. investigates the influence of two different types of vegetation (cushion plants and tussock grasses) on soil water balance, solute fluxes and chemical weathering in the high Andes ecosystem via fieldwork and numerical modelling. The cushion plants had lower water transmission below the A horizon and lower soil chemical weathering rates due to their shallower and coarser roots. The tussock grasses had steady water transmission throughout the soil profile and higher soil chemical weathering due to their deeper and finer roots. The paper concluded the soil water balance under the two vegetation types was different due to the root structure of the plants, which altered solute fluxes and soil chemical weathering throughout the depth of the soil.

Studies done in the past about the volcanic ash soils of the high Andes focused mostly on land use (Podwojewski et al., 2006; Buytaert et al., 2002; Buytaert et al. 2007) and on soil properties (Buytaert et al., 2005; Zehetner et al., 2003; Tonneijck et al., 2010; Tonneijck et al., 2010; Zehetner & Miller, 2006), while more recent studies in the area focused on its vegetation effects on hydraulic soil properties (Páez-Bimos et al., 2022), soil hydrology (Lahuatte et al., 2022), runoff processes (Minaya et al., 2021), and weathering (Barbosa et al., 2022). This paper is in line with these topics, since it focuses on vegetation, weathering, and hydrology. It is most comparable with Páez-Bimos et al. (2022) by also researching vegetation effects on hydraulic soil properties. Some of their conclusions were repeated in this paper, for example:

> - Compared to tussock grasses, the higher water retention capacity at saturation under cushion-forming plants can enhance soil water storage during prolonged rainfall events, whilst the higher total available water results in higher water storage for plants and can promote evapotranspiration during the dry season.

> - The saturated hydraulic conductivity in the top horizon is higher under cushion forming plants.

> - Below the rooting zone, the saturated hydraulic conductivity drops remarkably, especially under cushion plants.

However, I think there are sufficient differences between the two papers. Páez-Bimos et al. (2022) touches upon the soil pore structure and the specific plant root characteristics that were found to be notably different between the two vegetation types, which is not included in this paper. This paper focuses on solute fluxes and chemical weathering, which is not discussed in the paper by Páez-Bimos et al. (2022).

According to the paper, vegetation effects on individual components of the soil water balance have been studied before. Furthermore, Molina et al. (2019) found that there are significant differences in soil chemical weathering between vegetation patterns. This paper aims to reveal the mechanics

behind vegetation influencing soil weathering rates via its root system effects on soil water fluxes, since this has not been evidenced before. It adds knowledge about soil-vegetation-water interactions that may be relevant for soil hydrology, soil chemistry and ecology fields. Therefore, I think the paper addresses relevant scientific questions within the scope (advancing understanding of hydrological systems) of the journal Hydrology and Earth System Sciences well.

The paper is well written and uses understandable and precise language. It includes figures that support the understanding of, summarize, and add to the text well. Especially figure 10 helps to understand and summarize the text very well. The root diameter and deepness are visualized here, which puts their differences into perspective. The differences in arrow sizes are distinguishable and immediately show the results in a glance. However, I do have some issues – mostly with the methods and conclusions – that I would like the authors to address before publishing. Besides the strengths of the paper, I will elaborate on these issues below.

We thank you for the constructive comments and the detailed revision of our work, which is greatly appreciated.

I think your introduction is well written and gives a good background and understanding for the topic of the paper. It also includes relevant references. I think it creates a nice funnel from a broad perspective (the relationship between soil hydrology and chemical weathering and how this has been researched before; the relationship between vegetation, soil hydrology and weathering; and the Andes ecosystem) to the research questions and problem statement. My personal preference would be to move the information on the Páramo ecosystem (line 99-109) from the introduction to the site description in the methods (paragraph 3.1), but this is entirely up to your own preferences.

We will reconsider the organization of the introduction and see if the information on the páramo ecosystem is more appropriate in the description of the methods.

From the methods section in your paper, I understood almost everything that you did. Everything was written down very clearly and most already existing methods were referenced properly (e.g: the undisturbed samples by the multi-step apparatus (van Dam et al., 1994) in line 157-158; "... by the Eijkelkamp pressure membrane apparatus (Klute, 1986)" in line 160; and "... using the two heads method (5 and 10 cm) (Reynolds and Elrick, 1985)" in line 165-166. Aso, figures 1d and 1e helped visualize the vegetation types and table 1 helped understand the text by visualizing what these sample locations look like. However, the reasoning behind some of your methods are not fully clear to me. For example:

> - It is unclear from the text that you did not violate the assumption of independence between observations of the Mann Whitney U test since you took multiple samples from one location. Can you prove that you did not violate this assumption?

> This is a very valid point, and we will address it in the revised manuscript.

- Also, you write that the mean annual rainfall values (2019-2020) did not differ significantly between the two sites (line 375). You however do not show what statistical test is used to determine this significant difference. Can you include this in your methods?

Thanks for spotting this. We will rephrase this sentence.

- Furthermore, you write that, for preparing the soil samples for soil texture determinations, the carbonates and OM were removed. Bieganowski et al. (2018) writes that, while there is no uniform soil preparation standard compiled, the best known method describes that – besides OM – soluble and gypsum removal is obligatory, and besides carbonate, iron oxide removal is optional. Can you argue why you only removed the carbonate and OM and not these other components?

We used the standard procedure for grain size analyses with a Laser Diffraction Particle Size Analyser. This included separation of the fine earth by dry sieving (< 2mm), followed by sample treatment with demineralized water, with 10% HCl to remove carbonates and with 35% hydroperoxyde to remove organic material. Solubles are brought into solution by adding the demineralized water, and treating the samples in the ultrasound. We will provide additional information on the soil samples preparation for soil texture in L.167 – 170.

- In your introduction, you reference Molina et al. (2019): "Here, we take advantage of the mosaic-like distribution of vegetation types in the high Andes ecosystem, changing over short distances and allowing other factors (i.e., climate, geology, soil age, and topography) to remain constant". This sentence makes it seem like it is important for the topography to remain constant for this experiment. However, Molina et al. (2019) write "High Andean tropical ecosystems provide a good opportunity to study the association between chemical weathering, local topography, and vegetation patterns: the climate, parent material, and soil age can be held constant at the landscape scale, while the vegetation and slope morphology can vary greatly from the hilltops to the valley bottoms.". Therefore the topography can vary between the two locations. Another paper done at your study sites (Páez-Bimos et al., 2022) concludes "Soil hydraulic properties and soil pore structure changed in the uppermost horizons (A1 and A2) under cushion-forming plants and tussock grasses; whereas they did not change at topographic position.". Can you specify whether the topography between the two sites is the same and if it was not, whether this influences your conclusions?

In this paper, we selected two soil profiles that are located on the same topographic position, I.e. the summit position. In our previous work (Paez-Bimos et al., 2022), we analyzed the data for the entire toposequences, from summit to the valleys. We will further specify this in the revised manuscript.

The soil profiles are located in the same topographic position: summit or crest L. 128.

To conclude, while I think your scientific methods are clearly outlined and reproduceable. I think by elaborating more on these methods, your approach will have a more solid basis for the reader.

I like that you explain your results in the discussion with many references, e.g.: "The higher ETa under cushion-forming plants compared to tussock grasses is consistent with a recent study in the same sites based on water stable isotopes (Lahuatte et al., in rev.)" in line 483-484; and "The limitation of soil water transmission (e.g., by a reduction in vertical hydraulic conductivity) can result in saturation even at low rainfall intensities during prolonged precipitation events (Burt and Butcher, 1985). Thus, under tussock grass in the A horizon, the soil water storage capacity is limited by..." in line 497-499. However, your conclusion that a shallower and coarser root system is related to a more porous soil structure is not explained anywhere in the text. I think you should explain this in your discussion. I would recommend to reference to your previous paper Páez-Bimos (2022), possibly with some of the other results from there, like relating the strong decrease in saturated hydraulic conductivity with depth under cushion forming plants facilitating the soils to become saturated faster to the ash deposits.

Many thanks for this suggestion, we will revise this part of the discussion and make reference to the work in Páez-Bimos et al (2022) in the section 5.1

In your paper, you make some conclusions that I could not find significant evidence for in your paper. I think this weakens the conclusions of the paper: right now I am not convinced that the conclusions are substantiated. For example:

- In the conclusion (line 669), you write "Other solutes like DSi, Na, HCO3- are only minimally influenced by vegetation type.". However, the highest HCO3- concentration was found at the greatest depth under tussock grass which, according to the you, cannot be explained by the root network or SOC and needs further investigation (line 596). Therefore it is unclear to me how you reach the conclusion that there is only a minimal influence from vegetation type on HCO3- if the highest found concentration cannot be explained by the root network or SOC and needs further investigation. You now only include sources about HCO3- (in line 595) that found "higher root and microbial respiration can enhance carbonic acid formation, which then dissociates into HCO3-". Can you reference a source that finds the deep high HCO3- concentration cannot be explained by the root network or SOC?

This is a valid point, and we will rework this part.

- You conclude "In our study sites, we evidenced the role of root systems in regulating the soil water balance." (line 668-669). I did find statistical evidence that there were differences between the vegetation types in solute concentrations and fluxes, but I could not find statistical evidence that there were differences between vegetation types in the soil water balance or evidence for this causality. Can you explain how you proved the causality of vegetation root systems regulating the soil water balance?

We will rephrase this part, and making reference to the work in Paez-Bimos et al. (2022) where we show that there exist differences in soil water fluxes between the vegetation

types. The statistical methods do not allow us to prove causality, as there might be confounding factors.

- The annual solute fluxes of cations and DSi (taken as a proxy for weathering) are systematically higher in the TU-UP profile (line 449-451), but you write Na and DSi differences are not significant (line 589, although figure 8 shows Na differs significantly at 20 and 40 cm and DSi differs significantly at 40 cm) even though DSi was observed as one of the two dominant soil solute fluxes (line 538). Can you explain how you evidenced that there are chemical weathering differences between the vegetation types if this is the case?

L.449-451 refer to solute fluxes while L.589 refer to solute concentration. We will rephrase sentence in L.589.

- I cannot find statistical evidence for the causality relationship of the vegetation type affecting the contemporary soil chemical weathering (line 673-674). Can you prove that there is causality here?

This is a valid point, and we refer to our comment above. Causality cannot be proven with statistical inferences. We will rephrase this part of the Conclusions.

In conclusion, I do not think the statistics and results you showed are sufficient to support your conclusions at this moment. I think including some sources for HCO3- and evidence of causality will prove you reached substantial conclusions, which would improve the paper remarkably.

We will revise these parts based on your comments.

I also have some minor issues, that are summed up below:

- I am not sure if the title clearly reflects the contents of the paper. First of all, it does not become clear in the paper what the difference between solute flow and transport is. Also, the title is only associated with research question ii. There is no mention of chemical weathering, which was addressed in research question iii.

We will consider modify the title to include chemical weathering

- The abstract only mentions that the vertical distribution of soil properties associated with the root systems. It does not explain what the mechanism is here, while the role of the roots is explained in the conclusion quite well. I think this misses from the abstract. Also, the abstract does not mention the location of the measurements.

We will include both comments in the Abstract

- Your paper aims to fill the knowledge gap of how vegetation can influence contemporary weathering rates through its effect on soil water fluxes and transport. It is unclear how this information can be applied, can you give an argument for why the knowledge gap needs to be filled?

We will include an argument of the need for the research gap in the Intro

- In figure 4, the dots are very hard to distinguish from the simulated line. Also, the legend colors are very hard to relate to the colors in the picture.

We will modify the color of the lowest horizons to be more contrasting

- I think the number and quality of your references is appropriate. When I checked some of your sources (e.g. Fan et al., 2017; Jiang et al., 2018), I could not find any incorrect interpretations and these sources seemed of high scientific quality. However, I noticed that you included two papers in your reference list that were under revision by the time you wrote the paper. When I checked them it appeared that they have been published already. I will include them (Lahuatte et al., 2022; Páez-Bimos et al., 2022) in my reference list for you to include them.

We will update both references as they are published now

- Páez-Bimos (2022) quantifies the root abundance and diameter, is that what you used for the relative root sizes in figure 10? If so, can you reference this paper in your figure for that?

We will include the reference in Caption of Figure 10.

- The paper is inconsequent in the APA references in the text: when two authors are included, sometimes '&' is used, sometimes 'and' is used.

We will check the references as required for the HESS journal

- P7, line 187 and p9, line 235 reference to figure 2e, which does not exist.

We will change to 2d instead of 2e

- P7, line 160 writes ""36 undisturbed and 18 undisturbed samples". I think one of these should be "disturbed".

We will check the type of samples and correct

- P26, line 619 says "0.4 times, respectively". However, I am not sure this is correct since the two transmissions do not both seem 0.4. When I did my own calculation it seemed more like "0.44 and 0.79 times respectively"?

We will check and correct if necessary

References:

Barbosa, A. M., Francelino, M. R., Thomazini, A., Schaefer, C. E. G. R., Anjos, L. H. C., Pereira, M. G., & Lyra, G. B. (2022). The thermal regime and mineralogical attributes of highland volcanic-ash soils from the Cotopaxi volcano, Ecuador: Absent permafrost and little pedogenesis. Geoderma Regional, 29, e00496.

Bieganowski, A., Ryżak, M., Sochan, A., Barna, G., Hernádi, H., Beczek, M., ... & Makó, A. (2018). Laser diffractometry in the measurements of soil and sediment particle size distribution. Advances in agronomy, 151, 215-279.

Buytaert, W., Deckers, J., & Wyseure, G. (2007). Regional variability of volcanic ash soils in south Ecuador: The relation with parent material, climate and land use. Catena, 70(2), 143-154.

Buytaert, W., Deckers, J., Dercon, G., De Bievre, B., Poesen, J., & Govers, G. (2002). Impact of land use changes on the hydrological properties of volcanic ash soils in South Ecuador. Soil use and management, 18(2), 94-100.

Buytaert, W., Sevink, J., De Leeuw, B., & Deckers, J. (2005). Clay mineralogy of the soils in the south Ecuadorian páramo region. Geoderma, 127(1-2), 114-129.

Fan, Y., Miguez-Macho, G., Jobbágy, E. G., Jackson, R. B., and Otero-Casal, C.: Hydrologic regulation of plant rooting depth, Proc. Natl. Acad. Sci. U.S.A., 114, 10572–10577, https://doi.org/10.1073/pnas.1712381114, 2017.

Jiang, X. J., Liu, W., Chen, C., Liu, J., Yuan, Z.-Q., Jin, B., and Yu, X.: Effects of three morphometric features of roots on soil water flow behavior in three sites in China, Geoderma, 320, 161–171, https://doi.org/10.1016/j.geoderma.2018.01.035, 2018.

Lahuatte, B., Mosquera, G. M., Páez-Bimos, S., Calispa, M., Vanacker, V., Zapata-Ríos, X., ... & Crespo, P. (2022) Delineation of water flow paths in a tropical Andean headwater catchment with deep soils and permeable bedrock. Hydrological Processes, e14725.

Minaya, V., Camacho Suarez, V., Wenninger, J., & Mynett, A. (2021). Runoff generation from a combined glacier and páramo catchment within the Antisana Reserve in Ecuador. Journal of

Ecohydraulics, 1-16. Molina, A., Vanacker, V., Corre, M. D., and Veldkamp, E.: Patterns in soil chemical weathering related to topographic gradients and vegetation structure in a high Andean tropical ecosystem, Journal of Geophysical Research: Earth Surface, 124, 666–685, 2019.

Páez-Bimos, S., Villacís, M., Morales, O., Calispa, M., Molina, A., Salgado, S., ... & Vanacker, V. (2022). Vegetation effects on soil pore structure and hydraulic properties in volcanic ash soils of the high Andes. Hydrological Processes, 36(9), e14678.

Podwojewski, P., Poulenard, J., Zambrana, T., & Hofstede, R. (2002). Overgrazing effects on vegetation cover and properties of volcanic ash soil in the páramo of Llangahua and La Esperanza (Tungurahua, Ecuador). Soil Use and Management, 18(1), 45-55.

Tonneijck, F. H. (2006). Volcanic ash soils in Andean ecosystems: unravelling organic matter distribution and stabilisation. Radiocarbon, 48(3), 337-353.

Tonneijck, F. H., Jansen, B., Nierop, K. G. J., Verstraten, J. M., Sevink, J., & De Lange, L. (2010). Towards understanding of carbon stocks and stabilization in volcanic ash soils in natural Andean ecosystems of northern Ecuador. European Journal of Soil Science, 61(3), 392-405.

Zehetner, F., & Miller, W. P. (2006). Erodibility and runoff-infiltration characteristics of volcanic ash soils along an altitudinal climosequence in the Ecuadorian Andes. Catena, 65(3), 201-213.

Zehetner, F., Miller, W. P., & West, L. T. (2003). Pedogenesis of volcanic ash soils in Andean Ecuador. Soil Science Society of America Journal, 67(6), 1797-1809

---

## Author Response (AR1)

**Author comments (ACs)**

The authors thank the reviewers for their constructive comments. The comments are shown in black font, and our responses are in regular blue font. We indicated how we adjusted the manuscript in the part that is marked as "Correction:". Figure, page and line numbers in the reviewers' comments refer to the old manuscript, whereas they refer to the new manuscript in our responses.

**RC1: 'Comment on hess-2022-294', Anonymous Referee #1, 22 Oct 2022**

In general, I found the paper to be well written and the study well-designed and carried out.

I found – however – some points to be limited that needs to be addressed. First, the research gap was not clarified and the presented hypothesis was already addressed in other studies (see comments below). Some additional explanation on why the approach observation/modeling was used, would help the readers to capture the study design and idea early on in the work. I also found that the modeling was no explained with the necessary detail and should be accompanied by an uncertainty assessment.

Many thanks for the suggestions. We have strengthened the analyses by incorporating an uncertainty assessment, and provided more details.

Line 50 "the water balance"

Correction:
L.49: The word "the" has been included

L60: what about porosity and soil particle surface area?

We added a sentence based on same reference (Anderson et al., 2007)

Correction:
L.56-57: We added "Chemical weathering is conditioned by the intrinsic properties of the soil particles (e.g., porosity, soil particle surface area, mineralogy) and the soil solution (e.g., solution pH, conductivity, temperature) (Anderson et al., 2007)."

The break from L62 to L64 is quite harsh.

We added a sentence.

Correction:
L.63-64: Vegetation can directly control soil weathering depth by altering the composition of the soil solution but also indirectly by influencing soil hydrology (Kelly et al., 1998).

L89 and 90: We already know that the hypothesis is true. It has been shown in numerous studies before. The paper would greatly benefit to make the research gap more clear and have the

hypothesis and questions clearly linked to that. While the intro gives a good overview of what has been done, the research gap is only vaguely noted "Soil vegetation-water interactions are not fully understood". After that, the authors summarize things that are known, and not what is wrong in our current status quo. Points that are made e.g. in L116, should be in the introduction. This applies in general for section 2. Points that are made related to the research gap need to move into the intro.

We made the research gap clearer. While section 1 gives an overview of SoA and knowledge gaps in the field of soil ecohydrology and biogeochemical weathering, section 2 is more specific for the páramo ecosystem. We prefer to keep this structure, with section 2 focusing more on specificities of the páramo ecosystems

Correction:

L.70: we deleted "Soil-vegetation-water interactions are not fully understood."

L.88-94: we replaced "Here, we take advantage of the mosaic-like distribution of vegetation types in the high Andes ecosystem, changing over short distances and allowing other factors (i.e., climate, geology, soil age, and topography) to remain constant (Molina et al., 2019). We hypothesize that vegetation type has an impact on water and solute fluxes and further on soil chemical weathering at the soil profile scale."

with

"The effect of soil hydrology on chemical weathering has typically been studied indirectly through meteorological variables such as studies using long-term water balances based on the Budyko's framework (e.g. Calabrese and Porporato, 2020; Hunt, 2021). While such indirect assessments are useful for large-scale studies, they fail in capturing the variability in soil properties, topography, and vegetation patterns that may exist at small spatial scales (Calabrese et al., 2022; Li et al., 2013; Sullivan et al., 2022). Here, we address this research gap by taking advantage of the mosaic-like distribution of vegetation types in the high Andes ecosystem, changing over short distances and allowing other factors (i.e., climate, geology, soil age, and topography) to remain constant (Molina et al., 2019)."

An explanation on the methodological choices would be helpful, which type of measurement supports which question and for what is the modelling needed?

We re-wrote the end of the last paragraph in the intro that relates the measurement and modelling to the research questions.

Correction:

L.96-105: we replaced "The HYDRUS-1D model was used to simulate soil hydrological processes at daily timesteps, including evapotranspiration, deep drainage, and soil water storage. Simulated water fluxes at soil horizons were validated with independent field measurements. Soil solutions were sampled at biweekly intervals, and their compositions served to estimate solute fluxes. The influence of the infiltrated water 95 fluxes on solute fluxes and soil chemical weathering was analyzed"

with

"To analyze vegetation-soil associations in relation to the soil water balance, we used the HYDRUS-1D model to simulate soil hydrological processes including evapotranspiration, deep drainage, and soil water storage. Simulated soil moisture and water fluxes at soil horizons were calibrated and validated with independent field measurements. To analyze the influence of the infiltrated water fluxes on soil chemical weathering, we sampled soil solutions at biweekly intervals, and their compositions served to estimate solute fluxes. Overall, this study assesses the influence of vegetation type and associated soil properties on soil water balance, solute fluxes, and contemporary soil chemical weathering at the soil profile scale. Given that vegetation patterns in the High Andes are subject to rapid anthropogenic and/or climate change (Molina et al., 2015; Vanacker et al., 2018), this study also contributes to assess the potential impact of vegetation change on soil hydro-physical and chemical properties, soil water and nutrient balance, and leaching of soil solutes."

L123-125: not needed/relevant

L.129: We have deleted the first sentence "The páramo ecosystem supplies drinking water to more than half a million inhabitants of Ecuador's capital, Quito (EPMAPS and FONAG, 2018)."

L215: what are the sensitive parameters? Do you enforce that the relation between ksat from one depth to the other is retained? What parameter ranges were used for the inverse simulation? Why? In general, 3.4. lacks the necessary details to reproduce or understand the simulation setup.

We provided more details on the model parameters. Please note that we have done an uncertainty analysis, and we included the information for the sensitivity analysis in the text.

Correction: We added:

L.217-223: "The initial soil hydraulic parameters were defined from field measurements and/or previously fitted bimodal van Genuchten models (Páez-Bimos et al., 2022). To reduce the number of soil hydraulic parameters, we performed a global sensitivity analysis using the variance-based Sobol method (Sobol, 2001). The sensitivity analysis used parameter ranges defined from the field measurements ($\theta_S$, $\theta_r$, $K_{SAT}$; Páez-Bimos et al., 2022) or from the literature ($\alpha$, n, $w_2$, $\alpha_2$, $n_2$; Dettmann et al., 2014) for bimodal porous degraded organic soils. The tortuosity parameter ($\tau$) was set to 0.5 as proposed by Mualem, (1976) and recommended for soils with SOC < 18% (Dettmann et al., 2014). Table S4 in the supplementary materials contains the details on the sensitive parameters and the parameter ranges."

L.224: We added: "along with the initial parameters"

Páez-Bimos, S., Villacís, M., Morales, O., Calispa, M., Molina, A., Salgado, S., ... & Vanacker, V. (2022). Vegetation effects on soil pore structure and hydraulic properties in volcanic ash soils of the high Andes. Hydrological Processes, 36(9), e14678.

Dettmann, U., Bechtold, M., Frahm, E., & Tiemeyer, B. (2014). On the applicability of unimodal and bimodal van Genuchten–Mualem based models to peat and other organic soils under evaporation conditions. Journal of Hydrology, 515, 103-115.

Mualem, Y. (1976). A new model for predicting the hydraulic conductivity of unsaturated porous media. Water resources research, 12(3), 513-522.

Fig. 3. I guess the horizontal grey bars indicate the boundaries of a horizon. Yet, this needs to be explained in the captions and not left for guessing. CU-UR and TU-UP should also be explained in the caption or written out to make the figure stand-alone from the text.

Correction:
L.331-333: We have included the following text in the caption of Figure 3: "Horizontal grey bars indicate the boundaries between soil horizons. The information of the soil pits under cushion-forming plants (CU-UR) is plotted in green color, while the information from soils under tussock grasses (TU-UP) is plotted in orange".

L309ff. Do the authors have any idea on the general variation of these soil properties beyond the two profiles? I am wondering if the difference is random or if this is really an effect/linked to the vegetation. I understand that the sampling is laborious, but I guess we all know sites where ksat does change by several orders of magnitude on very small spatial differences. Even though I agree that the A horizons of the sites seem to be quite different.

Correction:
L.351-355: We included "We have observed the same vertical distribution of soil properties ($K_{SAT}$, $\theta_S$, $\theta_{FC}$, $\theta_{WP}$, $\theta_{TAW}$, BD, and SOC) and root characteristics in six other soil profiles of the same sub-catchment in Jatunhuayco (Páez-Bimos et al., 2022). Based on these observations, we assume that the differences that were observed between the two profiles (CU-UR and TU-UP) are indicative of the differences between soil profiles under cushion-forming plants and tussock grasses in similar topographic positions."

Páez-Bimos, S., Villacís, M., Morales, O., Calispa, M., Molina, A., Salgado, S., ... & Vanacker, V. (2022). Vegetation effects on soil pore structure and hydraulic properties in volcanic ash soils of the high Andes. Hydrological Processes, 36(9), e14678.

L327: What are the calibrated values and how do they differ between the profiles and to the measured values? What are the "sensitive parameters" (L.215).

We included in this section the sensitive parameters and the calibrated values. We compared them to the measured values.

Correction:
L.358-364: Sensitive parameters were identified from the initial 72 parameters (for each soil profile) by the Sobol method whereby parameter values varied by vegetation type (full details are given in

Supplementary Material 3). For the soil profile under cushion-forming plants, the most sensitive soil hydraulic parameters are the n parameters along depth; whereas for the tussock profile the most sensitive parameters are related to the soil hydraulic properties of the upper soil horizon (10 -25 cm, especially) (Table 2). The calibrated soil hydraulic parameters obtained from the inverse modelling are shown in Table 2. The standard deviations for most parameters are small indicating that the inverse modelling approach gave stable parameter estimates."

Table 2. Fitted soil hydraulic parameters (standard deviation)

| | CU-UR | | | TU-UP | |
|---|---|---|---|---|---|
| Parameter | Depth [cm] | Fitted value (1 SD) | Parameter | Depth [cm] | Fitted value (1 SD) |
| n [-] | 15 | 2.50 (0.09) | $\alpha$ [1 cm$^{-1}$] | 10 | 0.028 (0.0002) |
| n [-] | 25 | 1.21 (0.005) | n [-] | 10 | 2.50 (0.04) |
| n [-] | 35 | 2.50 (0.51) | $K_{SAT}$ [cm/d] | 10 | 4.96 (0.20) |
| n [-] | 45 | 1.23 (0.003) | $w_2$ [-] | 10 | 0.001 (0.0007) |
| n [-] | 55 | 2.50 (0.54) | $\alpha_2$ [1 cm$^{-1}$] | 10 | 0.007 (0.0008) |
| n [-] | 65 | 2.50 (0.61) | $n_2$ [-] | 10 | 1.50 (0.19) |
| n [-] | 85 | 1.26 (0.003) | $\alpha$ [1 cm$^{-1}$] | 25 | 0.003 (0.00004) |
| | | | n [-] | 25 | 2.23 (0.09) |
| | | | $\alpha$ [1 cm$^{-1}$] | 65 | 0.018 (0.0007) |
| | | | n [-] | 65 | 2.45 (0.10) |
| | | | $\alpha$ [1 cm$^{-1}$] | 75 | 0.004 (0.0003) |
| | | | n [-] | 75 | 2.50 (0.22) |

L333: I would partly disagree. When you have a KGE of 0.08, you barely explain anything of the observed behavior. So what is the problem? What could be the problem? Preferential flow?

Correction:
L.373-374: Despite the fact that the KGE values are generally low, they are above -0.41 indicating that the model adequately predicts the mean of the observations (Knoben et al., 2019).

Knoben, W. J., Freer, J. E., & Woods, R. A. (2019). Inherent benchmark or not? Comparing Nash–Sutcliffe and Kling–Gupta efficiency scores. Hydrology and Earth System Sciences, 23(10), 4323-4331.

Also, a full uncertainty analysis of the simulation should be added rather than 3 simulations in Figure 4. Furthermore, I am having a hard time to distinguish the different lines on the plot.

Thank you for this suggestion; we have now included a full uncertainty analysis.

Corrections:

1) We included the following paragraph in the subsection 3.4 (Water flux modeling):
L. 236-244: "The uncertainty on the modelled water fluxes was assessed using the Generalized Likelihood Uncertainty Estimation (GLUE) method (Beven and Binley, 1992). First, we selected the sensitive parameters, defined by the Sobol method, to randomly generate 50,000 parameter sets for each model (Selle et al., 2011; Kettridge et al., 2015). Second, we generated the sensitive parameters in the range of the fitted values plus or minus two standard deviations, as reported from

the inverse modelling. We assumed uniform distributions for all parameter sets. Third, we ran the models for the 50,000 parameter sets for the calibration and validation periods. We compared the simulated and observed soil moisture using the $R^2$ and KGE. We determined the behavioural parameter sets discarding simulations where $R^2 < 0.2$ and KGE < 0 (Houska et al., 2014). All model simulations, sensitivity analysis, uncertainty and calibration were carried out in the R programming language (R Development Core Team, 2010) by adapting the R packages: sensitivity and hydrusR (Acharya, 2020; Pujol et al., 2017)."

2) We included the following paragraph in the subsection 4.2 (Model simulations and independent validation):
L.375-377:" Figure 4 includes the range of uncertainty in the modelled volumetric water contents based on the 44,150 to 50,000 behavioural model runs for each soil horizon, and illustrates that the model performance decreases in the lower horizons.

3) We updated Figure 4 to include the uncertainty analysis and improved the visualization by making the lines thicker and the observation points a bit transparent.

L.398: We included in the caption "Colored lines show optimal model VWC estimate + 95% CI."

[Figure]

4) We included the following paragraph in the subsection 3.5.2 (Water chemical analysis):

L.295-301: "To propagate the uncertainty of the water fluxes to the solute fluxes, we selected randomly 10,000 behavioural model runs (from the total of 50,000 runs) of the biweekly water fluxes and used them in a linear regression model together with the available biweekly solute concentrations to complete the missing bi-weekly solute concentrations. We included the uncertainty of the linear regression by generating 100 random fitting parameters for each selected behavioural run. We determined $1 \times 10^6$ bi-weekly solute fluxes by multiplying the water fluxes by the solute concentrations. Finally, we aggregated the biweekly solute fluxes to annual values, and report the mean annual solute fluxes along with the 95% confidence intervals (mean ± 2 standard deviations)."

Beven K, Binley A. 1992. The future of distributed models: Model calibration and uncertainty prediction. Hydrological Processes 6 (3): 279–298 DOI: 10.1002/hyp.3360060305

Houska T, Multsch S, Kraft P. 2014. Monte Carlo-based calibration and uncertainty analysis of a coupled plant growth and hydrological model. Biogeosciences 11: 2069–2082

Kettridge N, Tilak AS, Devito KJ, Petrone RM, Mendoza CA, Waddington JM. 2015. Moss and peat hydraulic properties are optimized to maximize peatland water use efficiency. Ecohydrology 9 (6): 1039–1051 DOI: 10.1002/eco.1708

Selle B, Minasny B, Bethune M, Thayalakumaran T, Chandra S. 2011. Applicability of Richards' equation models to predict deep percolation under surface irrigation. Geoderma 160 (3–4): 569–578 DOI: 10.1016/j.geoderma.2010.11.005

5) We included the uncertainty of solute fluxes in Table 5
L.490-495: Table 5 incorporated mean and standard deviation

**Table 5:** Mean [and 95% confidence interval] of the annual solute fluxes [g $m^{-2}$ $y^{-1}$] for both study sites

| Soil profile | CU-UR | | | TU-UP | | |
|---|---|---|---|---|---|---|
| Horizon | A | 2A | 2BC | A | 2A | 3BC |
| Ca | 1.82 [0.96 − 2.68] | 1.26 [0.82 − 1.70] | 0.32 [0.22 − 0.42] | 6.20 [1.84 − 10.6] | 3.04 [1.60 − 4.48] | 2.40 [1.90 − 2.90] |
| Mg | 0.30 [0.18 − 0.42] | 0.38 [0.22 − 0.54] | 0.24 [0.16 − 0.32] | 1.97 [0.29 − 3.65] | 1.08 [0.04 −2.12] | 0.69 [0.19 − 1.19] |
| Na | 1.59 [1.29 − 1.89] | 1.62 [1.20 − 2.04] | 0.68 [0.52 − 0.84] | 2.52 [1.34 − 3.70] | 2.05 [1.33 − 2.77] | 2.63 [2.23 − 3.03] |
| K | 3.03 [1.95 − 4.11] | 0.17 [0.09 − 0.25] | 0.06 [0.02 − 0.10] | 1.36 [0.00 − 3.80] | 0.08 [0.06 − 0.10] | 0.07 [0.05 − 0.09] |
| $HCO_3^-$ | 11.1 [5.66 − 16.5] | 6.47 [3.57 − 9.37] | 2.70 [2.16 − 3.24] | 9.34 [4.44 − 14.2] | 10.3 [7.52 − 13.0] | 20.1 [16.1 − 24.1] |
| DOC | 17.4 [14.4 − 20.5] | 2.53 [1.77 − 3.29] | 0.36 [0.30 − 0.42] | 1.47 [1.23 − 1.71] | 1.58 [1.14 − 2.02] | 1.95 [1.51 − 2.39] |
| Al | 0.62 [0.40 − 0.84] | 0.04 [0.04 − 0.04] | 0.01 [0.01 − 0.01] | 0.03 [0.03- 0.03] | 0.03 [0.01 − 0.05] | 0.02 [0.00 − 0.04] |
| DSi | 5.95 [5.65 − 6.25] | 5.76 [4.14 − 7.38] | 3.29 [2.83 -3.75] | 7.25 [4.49 − 10.0] | 8.70 [7.16 − 10.2] | 9.61 [9.21 − 10.0] |
| T.Cat. | 11.1 [5.66 − 16.5] | 6.47 [3.57 − 9.37] | 2.70 [2.16 − 3.24] | 19.3 [13.2 − 25.5] | 15.0 [12.5 − 17.2] | 15.4 [13.1 − 17.7] |

T.Cat. = Sum of Ca, Mg, Na, K, and DSi fluxes
CI = confidence interval = mean ± 95% CI

L521: Replace "The soil hydrology' simulations" by "The simulation of the soil water balance"

Correction:
L.570: We have replaced ""The soil hydrology' simulations" with " The simulation of the soil water balance shows that surface runoff would not be"

L558: Can you estimate the residence time? Or the general difference between your sites?

Correction:
L.607: We have removed "a high water residence time attributed to"

L585ff. In this section, mostly literature is cited, however it would be more straightforward to argue from your observation rather than relying on a reference. Of course, other work can then be references.

We rewrote this section and argued directly from our observations.

Correction:
L.635fff:
We replaced

"Vegetation type has a minimal but not significant effect on Na and DSi fluxes (Fig. 9, Table 4). Annual DSi fluxes in our sites (4 - 9 g m$^{-2}$ y$^{-1}$) are within the range of DSi fluxes in other ecosystems (0 - 27 g m$^{-2}$ y$^{-1}$; Table S11). For cold environments low values of 0.04 to 1.1 g m$^{-2}$ y$^{-1}$were reported (Clow & Drever, 1996; Guicharnaud & Paton, 2006), whereas higher values up to 26 g m$^{-2}$ y$^{-1}$ were found for tropical warmer settings (Hedin et al., 2003). The solute fluxes of DSi and Na follow closely the variation of water fluxes with depth (Table 4, Table 3). Although not significant, the systematically higher DSi concentrations under cushion plants in comparison to tussock grass (Fig. 8d) can be a result of higher evapotranspiration and lower water fluxes as shown by White & Buss) (2014) and Berner & Berner (2012). The increase of DSi with depth under both vegetation types can be related to the longer water transit time at depth: (Lahuatte et al., in rev.) estimated water transit times for the A horizons at ~ 6 months, and for the 2A horizons at ~ 1 year. Bicarbonate concentrations seem to follow the vertical distribution of SOC content, root properties, and rooting depth, especially under cushion plants (Fig. 3d-3f, Fig. 8h). Higher root and microbial respiration can enhance carbonic acid formation, which then dissociates into HCO$_3^-$ (Amundson et al., 2007; Perdrial et al., 2015; Ugolini et al., 1988). The fact that we observed the highest concentration of HCO$_3^-$ at the greatest depth (3BC) under tussock grass cannot be explained by the root network or SOC and needs further investigation (Fig. 8h)."

with:

"Vegetation type has no significant effect on Na and DSi fluxes (Fig. 9, Table 5). Annual DSi fluxes in our sites (4 - 9 g $m^{-2}$ $y^{-1}$) are within the range of DSi fluxes in other ecosystems (0 - 27 g $m^{-2}$ $y^{-1}$; Table S11). The solute fluxes of DSi and Na follow closely the variation of water fluxes with depth (Table 5, Table 4). The increase of DSi with depth under both vegetation types can be related to the longer water transit time at depth: Lahuatte et al., (2022) estimated water transit times for the A horizons at ~ 6 months, and for the 2A horizons at ~ 1 year. Bicarbonate concentrations seem to follow the vertical distribution of SOC content, root properties, and rooting depth, especially under cushion plants (Fig. 3d-3f, Fig. 8h). Higher root and microbial respiration can enhance carbonic acid formation, which then dissociates into $HCO_3^-$ (Amundson et al., 2007; Perdrial et al., 2015; Ugolini et al., 1988). The fact that we observed the highest concentration of $HCO_3^-$ at the greatest depth (3BC) under tussock grass cannot be explained by the root network or SOC and needs further investigation (Fig. 8h)."

**RC2: 'Comment on hess-2022-294', Anonymous Referee #2, 15 Nov 2022**

This manuscript evaluates soil water and solute fluxes in two soil profiles under two different vegetation, i.e., the cushion-forming plants (CU-UR) and tussock grasses (TU-UP). This evaluation was based on measurements of soil water content, water flux density, and solute concentrations and the outputs from the well-known HYDRUS-1D model. These measurements and modeling allowed for evaluating the role of soil water balance on soil chemical weathering, which was one of the paper's main objectives.

Overall, the paper is well-written and suits the scope of HESS. Nevertheless, essential points must be included or further analyzed, especially in the methodology and discussion sections. These major points are described below, and a few minor considerations are described afterward.

We have addressed the comments, and have strengthened our work by including an uncertainty analysis, providing more details on methodological aspects, and giving more information on the determination of the Eta and ETp. We respond to your comments below in more detail:

In the introduction, the statement "we hypothesize that vegetation type has an impact on water and solute fluxes (...)'' does not suit well as a hypothesis since it has been well demonstrated, especially in the context of root water uptake modeling. Therefore, the hypothesis should be stated just as a particular case for these two vegetation types, not as an overall case.

Similar comment as RC1. We are not assessing root water uptake.

Correction:

L.88-94: we replaced "Here, we take advantage of the mosaic-like distribution of vegetation types in the high Andes ecosystem, changing over short distances and allowing other factors (i.e., climate, geology, soil age, and topography) to remain constant (Molina et al., 2019). We hypothesize that vegetation type has an impact on water and solute fluxes and further on soil chemical weathering at the soil profile scale."

with

"The effect of soil hydrology on chemical weathering has typically been studied indirectly through meteorological variables such as studies using long-term water balances based on the Budyko's framework (e.g. Calabrese and Porporato, 2020; Hunt, 2021). While such indirect assessments are useful for large-scale studies, they fail in capturing the variability in soil properties, topography, and vegetation patterns that may exist at small spatial scales (Calabrese et al., 2022; Li et al., 2013; Sullivan et al., 2022). Here, we address this research gap by taking advantage of the mosaic-like distribution of vegetation types in the high Andes ecosystem, changing over short distances and allowing other factors (i.e., climate, geology, soil age, and topography) to remain constant (Molina et al., 2019)."

There needs to be more information about how ETa and ETp were determined. Only lines 229-230 say that "ETa was derived from potential evapotranspiration (ETp) according to the surface pressure head and soil moisture'' and in the supplementary material 3 it is stated that "... ETp [was] based on the Penman-Monteith equation''. Notice that ETp from the Penman-Monteith method requires values of parameters such as crop canopy resistance and albedo. What were the values for each vegetation type? Furthermore, ETp is usually partitioned into potential transpiration and soil evaporation in hydrological modeling. Therefore, the authors should also shortly describe this and present the values of the parameters for each vegetation.

We included more details on the determination of the ETp and how we used the Penman-Monteith equation in this area. There are very few data on the plant transpiration and soil evaporation in these environments, and even less on the potential impact of vegetation roughness and albedo on evapotranspiration. The focus of this study is the soil water flux modelling, not that much the evapotranspiration modeling.

Correction:

L.249-255: "We calculated ETp based on the Penman-Monteith equation, as implemented in HYDRUS-1D (Šimůnek et al., 2018), using daily meteorological data from station JTU_AWS: incoming solar radiation, wind speed, relative humidity, as well as minimum and maximum air temperature. We left default values of meteorological parameters for cloudiness and emissivity on long wave radiation, as well as angstrom values for short wave radiation. We used 12 hours for daily sunshine and did not consider data for crops. The albedo was set to 0.14, which is the average of the albedo values that were reported earlier for the Ecuadorian páramo (0.11 – 0.17; (Montenegro-Díaz et al., 2022; Minaya et al., 2018)."

The methodology lacks information about root measurements, yet Figure 3 shows the vertical distribution of root diameter and abundance. Only the maximum rooting depth for each vegetation is given (lines 133--134). The authors also need to show how they determined the relative distribution of root length density over depth. How was it considered in the HYDRUS model? What about the transpiration reduction function? As far as I know, Hydrus-1D has two options for transpiration reduction functions: the Feddes and Jarvis model. I would like to know about values used for the empirical threshold parameters and if they can impact the simulation results.

We were not able to determined the root length density, and its variation over depth. It was originally foreseen to do more measurements on the root properties, but the restrictions during COVID-19 pandemic did not allow us to go to the field and the laboratory for further measurements. In the current model runs, we considered that the effect of the root system is included in the measured soil properties. As indicated above, we did not consider the partitioning of evapotranspiration into evaporation and transpiration. It was not the goal of this manuscript. We will provide more details on the root measurements in the revised manuscript.

Correction:

L.182-184: We included: "Plant root abundance and diameters were characterized in the field per genetic horizons following the procedures of the World Reference Base for Soil Resources (IUSS Working Group WRB, 2014)".

At the beginning of section 5.1, the authors compare ETa from CU-UR and TU-UP and cite values from other studies (in the second paragraph) but do not explain why ETa from CU-UR is higher than ETa from TU-UP. This difference is enhanced in dry periods but is also not explained. This discussion is brought back only in the fourth paragraph. First, in my opinion, this discussion should be placed right after the first paragraph.

Correction:

L.529-541: We placed the fourth paragraph after the first paragraph ad suggested.

Second, the authors show some evidence for the higher (lower) ETa from CU-UR (TU-UP) but need to address why this happens. For instance, the authors should explain why in 2A horizon under tussocks, which has about 50% of the roots (roughly looking at Figure 3f ), the highest soil moisture is observed, but annual ETa is well below ETp. Overall, the discussions are mainly based on soil water content, but when it comes to actual transpiration, one should look at the soil pressure head. Thus, the differences in soil hydraulic functions between soil layers and the two soil profiles need to be considered in the discussions. Papers related to root water uptake modeling might be useful to enhance these discussions.

We looked into the differences in the soil hydraulic functions between soil horizons and profiles, and analyze how they are related to differences in ETa and ETp.  We included in the discussion of section 5.1 the differences in fitted water retention curves. We are not evaluating root water uptake. See response below.

The discussion about soil water flux needs further analysis (section 5.1, 494--506). The authors attribute the differences in vertical water fluxes and deep drainage to the vertical distribution of soil water storage. However, notice that soil water content distribution is also affected by water flux. Also, stating that "soil water storage capacity is limited by lower θTAW  and KSAT'' is misleading since a high hydraulic conductivity promotes the reduction of soil water content in the soil layer. I think this discussion should be based on the soil hydraulic functions from each soil profile, considering the hydraulic conductivity and soil retention curve rather than on soil water content or soil water storage.

Many thanks for this suggestion; we included in the revised discussion of section 5.1 the differences in fitted soil hydraulic functions. We included also new Figures S8 and S9 in the Supplementary material 1 showing the water retention and hydraulic conductivity curves of the soil profiles.

Correction:
1) L.557-570: We re-wrote the paragraph:

"We attribute the differences in vertical water fluxes and deep drainage to the difference in the vertical distribution of soil hydraulic functions (Fig. S8, S9). In the A horizon near or above field capacity (pF ~ 2) high hydraulic conductivity and water retention under cushion-forming plants results in faster rainfall infiltration, higher water storage and higher evapotranspiration compared to under tussock grasses (Fig. S8a, S9a). This results in the dynamic range in soil moisture in the A horizon under cushion plants, which reflects the filling and emptying caused by low-intensity rainfall and evapotranspiration (Fig. 4a). Under cushion plants, the coarser root system and lower BD in the A horizon result in higher KSAT and water retention ($\theta$S, $\theta$FC) compared to tussock grass (Fig. 3c, 3e), as previously reported (Páez-Bimos et al., 2022). In the 2A horizon under tussock grass, the water retention near field capacity (pF ~ 2) is higher than in the A horizon as well as higher than under cushion plants in the 2A horizon (Fig. S8b). This results in higher soil moisture in the 2A horizon under tussock grass compared to the A horizon and the cushion plants in the 2A horizon (Fig. 4b). The hydraulic conductivity in the 2A horizon under both vegetation types lies in the same order of magnitude. In the 2BC horizon near field capacity, sol water retention for both vegetation types is the same range; however, the hydraulic conductivity is higher under tussock grass compared to under cushion-forming plants (Fig. S8c, S9c). The latter allows for a continued infiltration of water below this horizon under tussock grass."

2) New figures in supplementary material 1:

Figure S8. Water retention curves (matric potential vs. volumetric water content) plotted per horizon and soil profile, with CU-UR plotted in green and TU-UP plotted in orange. Two to three replica samples were analyzed per horizon and per profile.

[Figure]

Figure S9. Hydraulic conductivity curve (matric potential vs. hydraulic conductivity) plotted per horizon and soil profile, with CU-UR plotted in green and TU-UP plotted in orange. Two to three replica samples were analyzed per horizon and per profile.

[Figure]

Minor comments:

1) l187. There is no Figure 2e.

Correction:
L.191, 261: We 2e changed to 2d.

2) l218. The Kling-Gupta efficiency (KGE) may not be well known as the other measures for goodness-of-fit and may need some description and citation.

We included additional information on KGE

Correction:

L:228-230: we included "A KGE value greater than -0.41 indicates that the model predicts better than the mean of the observations, while a value of one indicates a perfect agreement between observed and simulated values (Knoben et al., 2019)."

3) l228. What do you mean by surface pressure head?

Correction:
L.249: "surface pressure head" replaced with "pressure head at the soil surface"

4) l334--336. These relations are not clear to me. Can we see them in any table?

Correction:

L.380: We include "(Table 2)".

L.366-368: **Table 2:** Fitted soil hydraulic parameters. The optimal model fit is given with 1 standard deviation (SD).

| CU-UR | | | TU-UP | | |
|---|---|---|---|---|---|
| Parameter | Depth [cm] | Fitted value (1 SD) | Parameter | Depth [cm] | Fitted value (1 SD) |
| n [-] | 15 | 2.50 (0.09) | $\alpha$ [1 cm$^{-1}$] | 10 | 0.028 (0.0002) |
| n [-] | 25 | 1.21 (0.005) | n [-] | 10 | 2.50 (0.04) |
| n [-] | 35 | 2.50 (0.51) | $K_{SAT}$ [cm/d] | 10 | 4.96 (0.20) |
| n [-] | 45 | 1.23 (0.003) | $w_2$ [-] | 10 | 0.001 (0.0007) |
| n [-] | 55 | 2.50 (0.54) | $\alpha_2$ [1 cm$^{-1}$] | 10 | 0.007 (0.0008) |
| n [-] | 65 | 2.50 (0.61) | $n_2$ [-] | 10 | 1.50 (0.19) |
| n [-] | 85 | 1.26 (0.003) | $\alpha$ [1 cm$^{-1}$] | 25 | 0.003 (0.00004) |
| | | | n [-] | 25 | 2.23 (0.09) |
| | | | $\alpha$ [1 cm$^{-1}$] | 65 | 0.018 (0.0007) |
| | | | n [-] | 65 | 2.45 (0.10) |
| | | | $\alpha$ [1 cm$^{-1}$] | 75 | 0.004 (0.0003) |
| | | | n [-] | 75 | 2.50 (0.22) |

5) l410. Measured or simulated values in Table 3?

We added "simulated" and "measured" in the Table 4 description

Correction:

L.459-460: **Table 4:** Mean annual simulated water fluxes and measured volumetric water content (Jan.-Dec.2019- Jan.-Dec.2020) by soil horizon (mean ± 2 standard deviations).

7) Fig 7. The line colors for each soil horizon are hard to recognize in the figure.

Correction:

We widen the width of the lines and change the color of the 3BC horizon.

[Figure]

6) l494. ``There is approximately 3-fold less water flux transmitted from the A to the underlying horizons under cushion plants (Table 3, Fig.     7b)''. It does not match to what is shown in Table 3.

Correction:

L.554: "There is approximately 2-fold less….."

7) The Equation 1 from the supplementary material 3 must have the sink term for root water uptake.

We did not include the sink term for root water uptake since we are not considering the partitioning of the evapotranspiration into evaporation and transpiration.

**CC1: 'Comment on hess-2022-294', Esther Geertsma, 07 Nov 2022**

The paper "Soil-vegetation-water interactions controlling solute flow and transport in volcanic ash soils of the high Andes" by Páez-Bimos et al. investigates the influence of two different types of vegetation (cushion plants and tussock grasses) on soil water balance, solute fluxes and chemical weathering in the high Andes ecosystem via fieldwork and numerical modelling. The cushion plants had lower water transmission below the A horizon and lower soil chemical weathering rates due to their shallower and coarser roots. The tussock grasses had steady water transmission throughout the soil profile and higher soil chemical weathering due to their deeper and finer roots. The paper concluded the soil water balance under the two vegetation types was different due to the root structure of the plants, which altered solute fluxes and soil chemical weathering throughout the depth of the soil.

Studies done in the past about the volcanic ash soils of the high Andes focused mostly on land use (Podwojewski et al., 2006; Buytaert et al., 2002; Buytaert et al. 2007) and on soil properties (Buytaert et al., 2005; Zehetner et al., 2003; Tonneijck et al., 2010; Tonneijck et al., 2010; Zehetner & Miller, 2006), while more recent studies in the area focused on its vegetation effects on hydraulic soil properties (Páez-Bimos et al., 2022), soil hydrology (Lahuatte et al., 2022), runoff processes (Minaya et al., 2021), and weathering (Barbosa et al., 2022). This paper is in line with these topics, since it focuses on vegetation, weathering, and hydrology. It is most comparable with Páez-Bimos et al. (2022) by also researching vegetation effects on hydraulic soil properties. Some of their conclusions were repeated in this paper, for example:

> - Compared to tussock grasses, the higher water retention capacity at saturation under cushion-forming plants can enhance soil water storage during prolonged rainfall events, whilst the higher total available water results in higher water storage for plants and can promote evapotranspiration during the dry season.

> - The saturated hydraulic conductivity in the top horizon is higher under cushion forming plants.

> - Below the rooting zone, the saturated hydraulic conductivity drops remarkably, especially under cushion plants.

However, I think there are sufficient differences between the two papers. Páez-Bimos et al. (2022) touches upon the soil pore structure and the specific plant root characteristics that were found to be notably different between the two vegetation types, which is not included in this paper. This paper focuses on solute fluxes and chemical weathering, which is not discussed in the paper by Páez-Bimos et al. (2022).

According to the paper, vegetation effects on individual components of the soil water balance have been studied before. Furthermore, Molina et al. (2019) found that there are significant differences in soil chemical weathering between vegetation patterns. This paper aims to reveal the mechanics behind vegetation influencing soil weathering rates via its root system effects on soil water fluxes,

since this has not been evidenced before. It adds knowledge about soil-vegetation-water interactions that may be relevant for soil hydrology, soil chemistry and ecology fields. Therefore, I think the paper addresses relevant scientific questions within the scope (advancing understanding of hydrological systems) of the journal Hydrology and Earth System Sciences well.

The paper is well written and uses understandable and precise language. It includes figures that support the understanding of, summarize, and add to the text well. Especially figure 10 helps to understand and summarize the text very well. The root diameter and deepness are visualized here, which puts their differences into perspective. The differences in arrow sizes are distinguishable and immediately show the results in a glance. However, I do have some issues – mostly with the methods and conclusions – that I would like the authors to address before publishing. Besides the strengths of the paper, I will elaborate on these issues below.

We thank you for the constructive comments and the detailed revision of our work, which is greatly appreciated.

I think your introduction is well written and gives a good background and understanding for the topic of the paper. It also includes relevant references. I think it creates a nice funnel from a broad perspective (the relationship between soil hydrology and chemical weathering and how this has been researched before; the relationship between vegetation, soil hydrology and weathering; and the Andes ecosystem) to the research questions and problem statement. My personal preference would be to move the information on the Páramo ecosystem (line 99-109) from the introduction to the site description in the methods (paragraph 3.1), but this is entirely up to your own preferences.

We made the research gap clearer L.88-94 (see response to RC1). While section 1 gives an overview of SoA and knowledge gaps in the field of soil ecohydrology and biogeochemical weathering, section 2 is more specific for the páramo ecosystem. We prefer to keep this structure, with section 2 focusing more on specificities of the páramo ecosystems.

From the methods section in your paper, I understood almost everything that you did. Everything was written down very clearly and most already existing methods were referenced properly (e.g: the undisturbed samples by the multi-step apparatus (van Dam et al., 1994) in line 157-158; "... by the Eijkelkamp pressure membrane apparatus (Klute, 1986)" in line 160; and "... using the two heads method (5 and 10 cm) (Reynolds and Elrick, 1985)" in line 165-166. Aso, figures 1d and 1e helped visualize the vegetation types and table 1 helped understand the text by visualizing what these sample locations look like. However, the reasoning behind some of your methods are not fully clear to me. For example:

> - It is unclear from the text that you did not violate the assumption of independence between observations of the Mann Whitney U test since you took multiple samples from one location. Can you prove that you did not violate this assumption?
>
> We used the Mann-Whitney U test to test if the solute concentrations and fluxes are significantly different ($p < 0.05$) between two groups or vegetation types: cushion-forming plants vs. tussock grasses). We fulfill the assumption of independence, since the measurements of solute concentrations and fluxes in each group (cushion-forming plants

vs. tussock grasses) are independent from each other.  Moreover, the Mann-Whitney U test is suited for small non-normal distributed datasets as evidenced by the Shapiro-Wilk test (p < 0.05). We re-wrote a part of the paragraph.

Correction:

L.304-305: "The Mann-Whitney U test was applied to test for significant differences (p < 0.05) in solute concentrations and solute fluxes between two groups (vegetation types: cushion-forming plants vs. tussock grasses)."

L.306-307:"The Mann-Whitney U test is suited for small non-normal distributed datasets as evidenced by the Shapiro-Wilk test (p < 0.05)."

- Also, you write that the mean annual rainfall values (2019-2020) did not differ significantly between the two sites (line 375). You however do not show what statistical test is used to determine this significant difference. Can you include this in your methods?

Correction:

We removed "significantly" in L.419 to avoid confusion.

- Furthermore, you write that, for preparing the soil samples for soil texture determinations, the carbonates and OM were removed. Bieganowski et al. (2018) writes that, while there is no uniform soil preparation standard compiled, the best known method describes that – besides OM – soluble and gypsum removal is obligatory, and besides carbonate, iron oxide removal is optional. Can you argue why you only removed the carbonate and OM and not these other components?

We used the standard procedure for grain size analyses with a Laser Diffraction Particle Size Analyser. This included separation of the fine earth by dry sieving (< 2mm), followed by sample treatment with demineralized water, with 10% HCl to remove carbonates and with 35% hydroperoxyde to remove organic material. Solubles and gypsum are removed as the samples are brought into solution by adding the demineralized water. Samples were treated with ultasonics to disperse clays. We provided additional information on the soil samples preparation for soil texture.

Correction:

L.171-172:  included "removal of solutes and gypsum (if any) with demineralized water,"

- In your introduction, you reference Molina et al. (2019): "Here, we take advantage of the mosaic-like distribution of vegetation types in the high Andes ecosystem, changing over short distances and allowing other factors (i.e., climate, geology, soil age, and topography) to remain constant". This sentence makes it seem like it is important for the topography to remain constant for this experiment. However, Molina et al. (2019) write "High Andean tropical ecosystems provide a good opportunity to study the association between chemical weathering, local topography, and vegetation patterns: the climate, parent material, and soil age can be held constant at the landscape scale, while the vegetation and slope

morphology can vary greatly from the hilltops to the valley bottoms.". Therefore the topography can vary between the two locations. Another paper done at your study sites (Páez-Bimos et al., 2022) concludes "Soil hydraulic properties and soil pore structure changed in the uppermost horizons (A1 and A2) under cushion-forming plants and tussock grasses; whereas they did not change at topographic position.". Can you specify whether the topography between the two sites is the same and if it was not, whether this influences your conclusions?

In this paper, we selected two soil profiles that are located on the same topographic position, i.e. the summit position. In our previous work (Paez-Bimos et al., 2022), we analyzed the data for the entire toposequences, from summit to the valleys.

Correction:

L.132: We included "at the summit topographic position"

To conclude, while I think your scientific methods are clearly outlined and reproduceable. I think by elaborating more on these methods, your approach will have a more solid basis for the reader.

I like that you explain your results in the discussion with many references, e.g.: "The higher ETa under cushion-forming plants compared to tussock grasses is consistent with a recent study in the same sites based on water stable isotopes (Lahuatte et al., in rev.)" in line 483-484; and "The limitation of soil water transmission (e.g., by a reduction in vertical hydraulic conductivity) can result in saturation even at low rainfall intensities during prolonged precipitation events (Burt and Butcher, 1985). Thus, under tussock grass in the A horizon, the soil water storage capacity is limited by..." in line 497-499. However, your conclusion that a shallower and coarser root system is related to a more porous soil structure is not explained anywhere in the text. I think you should explain this in your discussion. I would recommend to reference to your previous paper Páez-Bimos (2022), possibly with some of the other results from there, like relating the strong decrease in saturated hydraulic conductivity with depth under cushion forming plants facilitating the soils to become saturated faster to the ash deposits.

We revised made reference to the work in Páez-Bimos et al (2022) in the section 5.1. See also response to RC2.

Correction:

L.557-570: We re-wrote the paragraph:

"We attribute the differences in vertical water fluxes and deep drainage to the difference in the vertical distribution of soil hydraulic functions (Fig. S8, S9). In the A horizon near or above field capacity (pF ~ 2) high hydraulic conductivity and water retention under cushion-forming plants results in faster rainfall infiltration, higher water storage and higher evapotranspiration compared to under tussock grasses (Fig. S8a, S9a). This results in the dynamic range in soil moisture in the A horizon under cushion plants, which reflects the filling and emptying caused by low-intensity rainfall and evapotranspiration (Fig. 4a). Under cushion plants, the coarser root system and lower BD in the A horizon result in higher $K_{SAT}$ and water retention ($\theta_S$, $\theta_{FC}$) compared to tussock grass (Fig. 3c,

3e), as previously reported (Páez-Bimos et al., 2022). In the 2A horizon under tussock grass, the water retention near field capacity (pF ~ 2) is higher than in the A horizon as well as higher than under cushion plants in the 2A horizon (Fig. S8b). This results in higher soil moisture in the 2A horizon under tussock grass compared to the A horizon and the cushion plants in the 2A horizon (Fig. 4b). The hydraulic conductivity in the 2A horizon under both vegetation types lies in the same order of magnitude. In the 2BC horizon near field capacity, soil water retention for both vegetation types is the same range; however, the hydraulic conductivity is higher under tussock grass compared to under cushion-forming plants (Fig. S8c, S9c). The latter allows for a continued infiltration of water fluxes below this horizon under tussock grass."

In your paper, you make some conclusions that I could not find significant evidence for in your paper. I think this weakens the conclusions of the paper: right now I am not convinced that the conclusions are substantiated. For example:

- In the conclusion (line 669), you write "Other solutes like DSi, Na, HCO3- are only minimally influenced by vegetation type.". However, the highest HCO3- concentration was found at the greatest depth under tussock grass which, according to the you, cannot be explained by the root network or SOC and needs further investigation (line 596). Therefore it is unclear to me how you reach the conclusion that there is only a minimal influence from vegetation type on HCO3- if the highest found concentration cannot be explained by the root network or SOC and needs further investigation. You now only include sources about HCO3- (in line 595) that found "higher root and microbial respiration can enhance carbonic acid formation, which then dissociates into HCO3-". Can you reference a source that finds the deep high HCO3- concentration cannot be explained by the root network or SOC?

We agree that we cannot make a conclusive statement about the $HCO_3^-$ concentrations, as the depth-variation in $HCO_3^-$ concentrations under tussock grasses needs to be further investigated. Therefore, in the conclusions, we have removed the reference to "$HCO_3^-$".

Correction:

L.715-716: "Other solutes like DSi, and Na are only minimally influenced by vegetation type."

- You conclude "In our study sites, we evidenced the role of root systems in regulating the soil water balance." (line 668-669). I did find statistical evidence that there were differences between the vegetation types in solute concentrations and fluxes, but I could not find statistical evidence that there were differences between vegetation types in the soil water balance or evidence for this causality. Can you explain how you proved the causality of vegetation root systems regulating the soil water balance?

We rephrased this part, as you are correctly noting down that this was not statistically tested. The number of observations is not high enough to do statistical analyses on these data.

Correction:

L. 704-705: "In our study sites, we found associations between root systems (related to vegetation type) and soil water balance and fluxes."

- The annual solute fluxes of cations and DSi (taken as a proxy for weathering) are systematically higher in the TU-UP profile (line 449-451), but you write Na and DSi differences are not significant (line 589, although figure 8 shows Na differs significantly at 20 and 40 cm and DSi differs significantly at 40 cm) even though DSi was observed as one of the two dominant soil solute fluxes (line 538). Can you explain how you evidenced that there are chemical weathering differences between the vegetation types if this is the case?

You are correct, this was confusing. We rephrased the sentence

Correction:

L.635: "Vegetation type has a minimal but not significant effect on Na and DSi fluxes…"

- I cannot find statistical evidence for the causality relationship of the vegetation type affecting the contemporary soil chemical weathering (line 673-674). Can you prove that there is causality here?

We rephrased this part of the Conclusions.

Correction:

L. 719-720: "contemporary soil chemical weathering rates differ by vegetation type, as the vegetation  modifies the soil hydraulic properties in the upper horizon,…"

In conclusion, I do not think the statistics and results you showed are sufficient to support your conclusions at this moment. I think including some sources for HCO3- and evidence of causality will prove you reached substantial conclusions, which would improve the paper remarkably.

We have revised parts of the manuscript based on your comments. See above.

I also have some minor issues, that are summed up below:

- I am not sure if the title clearly reflects the contents of the paper. First of all, it does not become clear in the paper what the difference between solute flow and transport is. Also, the title is only associated with research question ii. There is no mention of chemical weathering, which was addressed in research question iii.

Correction:

L.2: Title changed from "Soil-vegetation-water interactions controlling solute flow and transport in volcanic ash soils of the high Andes" to "Soil-vegetation-water interactions controlling solute flow and chemical weathering in volcanic ash soils of the high Andes"

- The abstract only mentions that the vertical distribution of soil properties associated with the root systems. It does not explain what the mechanism is here, while the role of the roots is explained in the conclusion quite well. I think this misses from the abstract. Also, the abstract does not mention the location of the measurements.

Correction:

1) L.22-23: We changed:

"This is attributed to the vertical distribution of soil properties associated with the root systems" with:

"This is attributed to the higher soil water retention and saturated hydraulic conductivity associated with a shallower and coarser root system."

2) L.17-18: We included "in the high Ecuadorian Andes"

- Your paper aims to fill the knowledge gap of how vegetation can influence contemporary weathering rates through its effect on soil water fluxes and transport. It is unclear how this information can be applied, can you give an argument for why the knowledge gap needs to be filled?

We included an argument at the end of the introduction.

Correction:

L.103-105: Given that vegetation patterns in the High Andes are subject to rapid anthropogenic and/or climate change (Molina et al., 2015; Vanacker et al., 2018), this study also contributes to assess the potential impact of vegetation change on soil hydro-physical and chemical properties, soil water and nutrient balance, and leaching of soil solutes.

- In figure 4, the dots are very hard to distinguish from the simulated line. Also, the legend colors are very hard to relate to the colors in the picture.

Correction:

We made the lines thicker and the points a bit transparent.

- I think the number and quality of your references is appropriate. When I checked some of your sources (e.g. Fan et al., 2017; Jiang et al., 2018), I could not find any incorrect interpretations and these sources seemed of high scientific quality. However, I noticed that you included two papers in your reference list that were under revision by the time you wrote the paper. When I checked them it appeared that they have been published already. I will include them (Lahuatte et al., 2022; Páez-Bimos et al., 2022) in my reference list for you to include them.

Correction:

We have updated both references in the manuscript.

- Páez-Bimos (2022) quantifies the root abundance and diameter, is that what you used for the relative root sizes in figure 10? If so, can you reference this paper in your figure for that?

Correction:

L.694-695: We added: "Root depth, abundance and diameter drawn as per Páez-Bimos et al., (2022)."

- The paper is inconsequent in the APA references in the text: when two authors are included, sometimes '&' is used, sometimes 'and' is used.

Correction:

We checked and corrected all references from "&" to "and" as required for the HESS journal

- P7, line 187 and p9, line 235 reference to figure 2e, which does not exist.

Correction:
L.191, 261: We 2e changed to 2d. (same comment as RC2)

- P7, line 160 writes ""36 undisturbed and 18 undisturbed samples". I think one of these should be "disturbed".

Correction:

L.162: changed "18 undisturbed" to "18 disturbed"

- P26, line 619 says "0.4 times, respectively". However, I am not sure this is correct since the two transmissions do not both seem 0.4. When I did my own calculation it seemed more like "0.44 and 0.79 times respectively"?

Correction:

L.666: we changed "0.4 times, respectively" to "0.4 and 0.3 times, respectively"

References:

Barbosa, A. M., Francelino, M. R., Thomazini, A., Schaefer, C. E. G. R., Anjos, L. H. C., Pereira, M. G., & Lyra, G. B. (2022). The thermal regime and mineralogical attributes of highland volcanic-ash soils from the Cotopaxi volcano, Ecuador: Absent permafrost and little pedogenesis. Geoderma Regional, 29, e00496.

Bieganowski, A., Ryżak, M., Sochan, A., Barna, G., Hernádi, H., Beczek, M., ... & Makó, A. (2018). Laser diffractometry in the measurements of soil and sediment particle size distribution. Advances in agronomy, 151, 215-279.

Buytaert, W., Deckers, J., & Wyseure, G. (2007). Regional variability of volcanic ash soils in south Ecuador: The relation with parent material, climate and land use. Catena, 70(2), 143-154.

Buytaert, W., Deckers, J., Dercon, G., De Bievre, B., Poesen, J., & Govers, G. (2002). Impact of land use changes on the hydrological properties of volcanic ash soils in South Ecuador. Soil use and management, 18(2), 94-100.

Buytaert, W., Sevink, J., De Leeuw, B., & Deckers, J. (2005). Clay mineralogy of the soils in the south Ecuadorian páramo region. Geoderma, 127(1-2), 114-129.

Fan, Y., Miguez-Macho, G., Jobbágy, E. G., Jackson, R. B., and Otero-Casal, C.: Hydrologic regulation of plant rooting depth, Proc. Natl. Acad. Sci. U.S.A., 114, 10572–10577, https://doi.org/10.1073/pnas.1712381114, 2017.

Jiang, X. J., Liu, W., Chen, C., Liu, J., Yuan, Z.-Q., Jin, B., and Yu, X.: Effects of three morphometric features of roots on soil water flow behavior in three sites in China, Geoderma, 320, 161–171, https://doi.org/10.1016/j.geoderma.2018.01.035, 2018.

Lahuatte, B., Mosquera, G. M., Páez-Bimos, S., Calispa, M., Vanacker, V., Zapata-Ríos, X., ... & Crespo, P. (2022) Delineation of water flow paths in a tropical Andean headwater catchment with deep soils and permeable bedrock. Hydrological Processes, e14725.

Minaya, V., Camacho Suarez, V., Wenninger, J., & Mynett, A. (2021). Runoff generation from a combined glacier and páramo catchment within the Antisana Reserve in Ecuador. Journal ofEcohydraulics, 1-16.

Molina, A., Vanacker, V., Brisson, E., Mora, D., and Balthazar, V.: Multidecadal change in streamflow associated with anthropogenic disturbances in the tropical Andes, Hydrol. Earth Syst. Sci., 19, 4201–4213, https://doi.org/10.5194/hess-19-4201-2015, 2015.

Molina, A., Vanacker, V., Corre, M. D., and Veldkamp, E.: Patterns in soil chemical weathering related to topographic gradients and vegetation structure in a high Andean tropical ecosystem, Journal of Geophysical Research: Earth Surface, 124, 666–685, 2019.

Páez-Bimos, S., Villacís, M., Morales, O., Calispa, M., Molina, A., Salgado, S., ... & Vanacker, V. (2022). Vegetation effects on soil pore structure and hydraulic properties in volcanic ash soils of the high Andes. Hydrological Processes, 36(9), e14678.

Podwojewski, P., Poulenard, J., Zambrana, T., & Hofstede, R. (2002). Overgrazing effects on vegetation cover and properties of volcanic ash soil in the páramo of Llangahua and La Esperanza (Tungurahua, Ecuador). Soil Use and Management, 18(1), 45-55.

Tonneijck, F. H. (2006). Volcanic ash soils in Andean ecosystems: unravelling organic matter distribution and stabilisation. Radiocarbon, 48(3), 337-353.

Tonneijck, F. H., Jansen, B., Nierop, K. G. J., Verstraten, J. M., Sevink, J., & De Lange, L. (2010). Towards understanding of carbon stocks and stabilization in volcanic ash soils in natural Andean ecosystems of northern Ecuador. European Journal of Soil Science, 61(3), 392-405.

Veerle Vanacker, Armando Molina, Rossana Torres, Edison Calderon & Laura Cadilhac (2018) Challenges for research on global change in mainland Ecuador, Neotropical Biodiversity, 4:1, 114-118, DOI: 10.1080/23766808.2018.1491706

Zehetner, F., & Miller, W. P. (2006). Erodibility and runoff-infiltration characteristics of volcanic ash soils along an altitudinal climosequence in the Ecuadorian Andes. Catena, 65(3), 201-213.

Zehetner, F., Miller, W. P., & West, L. T. (2003). Pedogenesis of volcanic ash soils in Andean Ecuador. Soil Science Society of America Journal, 67(6), 1797-1809